# TAU-106K: A New Dataset for Comprehensive Understanding of Traffic Accident

**Yixuan Zhou**[1,*] **Long Bai**[1,*] **Sijia Cai**[1,†,‡] **Bing Deng**[1] **Xing Xu**[2,‡] **Heng Tao Shen**[2]

[1] Alibaba Cloud Computing  [2] Tongji University (Tongji)

{yixuanzhou.zyx, bailong.bai, stephen.csj, dengbing.db}@alibaba-inc.com,
interxuxing@hotmail.com, shenhengtao@hotmail.com

## Abstract

Multimodal Large Language Models (MLLMs) have demonstrated impressive performance in general visual understanding tasks. However, their potential for high-level, fine-grained comprehension, such as anomaly understanding, remains unexplored. Focusing on traffic accidents, a critical and practical scenario within anomaly understanding, we investigate the advanced capabilities of MLLMs and propose TABot, a multimodal MLLM specialized for accident-related tasks. To facilitate this, we first construct TAU-106K, a large-scale multimodal dataset containing 106K traffic accident videos and images collected from academic benchmarks and public platforms. The dataset is meticulously annotated through a video-to-image annotation pipeline to ensure comprehensive and high-quality labels. Building upon TAU-106K, we train TABot using a two-step approach designed to integrate multi-granularity tasks, including accident recognition, spatial-temporal grounding, and an auxiliary description task to enhance the model's understanding of accident elements. Extensive experiments demonstrate TABot's superior performance in traffic accident understanding, highlighting not only its capabilities in high-level anomaly comprehension but also the robustness of the TAU-106K benchmark. Our code and data will be available at https://github.com/cool-xuan/TABot.

## 1 Introduction

Traffic Accident Detection (TAD) has always been a crucial and practical task in public safety and transportation management. The development of advanced technologies, such as computer vision and deep learning, has enabled the automation of TAD, providing real-time accident alerts and analysis. Despite significant research on traffic accident detection (Shah et al., 2018; Zhu et al., 2019; Kim et al., 2019; Haresh et al., 2020; Ghahremannezhad et al., 2022), existing TAD methods often rely on conventional visual feature extraction techniques that lack high-level comprehension and multimodal alignment capabilities in interpreting intricate spatial-temporal dynamics.

Recent advancements in aligning large language models (LLMs) on visual-language datasets have driven remarkable progress in multi-modal understanding tasks (Achiam et al., 2023). With extensive pre-training and instruction tuning, multimodal LLMs (MLLMs) have demonstrated superior performance in spatially visual understanding and logical reasoning for the general-purpose comprehension (Li et al., 2023; Zhu et al., 2023; Liu et al., 2024). However, due to the inherent gap between general and specific tasks, MLLMs trained on general domains often underperform in specialized fields due to limited domain-specific data. For instance, as illustrated in Figure 1, the most advanced MLLMs, such as GPT-4o (Achiam et al., 2023), Qwen2-VL (Yang et al., 2024), and Gemini-1.5-Pro (Team et al., 2024), may misinterpret the visual content of traffic accidents where

---

*Equal contribution. Work done during an internship at Alibaba Cloud Computing.
†Project lead.
‡Corresponding authors.

> **Question:** Do you notice any traffic accident scenarios in this picture? Please respond with a 'Yes' or 'No'. Following that, describe the image in one sentence.
>
> 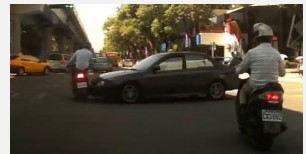
>
> **GPT-4o:** No, the image shows a busy city intersection with vehicles and motorcycles waiting at a traffic light.
> **Qwen2-VL:** No. The image shows a busy urban street with cars, scooters, and pedestrians, but there is no visible accident.
> **Gemini-1.5-Pro:** No. The image shows a sunny city street scene with several motorbikes and cars at an intersection under a highway overpass.
> **TABot-Chat (Ours):** Yes. Because a black car brake was not on time, it collided with a white coat electric bicycle rider.

Figure 1: One example to illustrate the limitations of general MLLM in understanding traffic accidents. In the scenario where a vehicle collision occurs due to a sudden lane change by the leading vehicle, **GPT-4o**, **Qwen2-VL**, and **Gemini-1.5-Pro** *fail* to detect this issue.

a vehicle collision occurs, leading to inaccurate responses. We argue that the failure of general MLLMs to understand traffic accidents stems from the following two main reasons: (i) Traffic accident detection requires MLLMs to grasp ambiguous concepts like *anomaly* and *accident*, which are context-dependent and defined by human criteria. However, existing MLLMs are trained on general-purpose data focusing on fundamental semantics, lacking the specialized understanding of such high-level semantics. (ii) The visual representations of accident occurrences differ significantly from general scenes, necessitating realigning these visual representations with the semantic understanding towards traffic accidents. Both of these limitations highlight the need for infrastructure that includes accident-specific annotations and specialized MLLMs to understand traffic accidents.

To pioneer an MLLM specialized in traffic accident comprehension, we first created **TAU-106K**, a large-scale multimodal traffic accident dataset containing 106K videos and images with detailed accident-oriented annotations. In particular, we aggregate academic benchmarks and crawl traffic accident videos from public platforms, building a diverse and high-quality visual foundation. To ensure annotation quality and efficiency, we design a video-to-image annotation pipeline, resulting in comprehensive annotations that are manually crafted by human labors. The annotations cover accident recognition, description, temporal localization, and spatial grounding at both the image and video levels, providing detailed and structured information for MLLMs to understand traffic accidents.

Using TAU-106K, we reorganize the annotations into instructional data to unlock MLLMs' potential in traffic accident understanding and introduce **TABot**, an end-to-end MLLM specialized for traffic accident comprehension across both image and video modalities. We adopt a two-step training approach: functional tuning to engage multi-granularity accident detection capabilities activation, and instruction tuning to enhance contextual comprehension and instruction following capabilities. In particular, during functional tuning, we propose two training strategies to serve temporal localization, the most crucial task in traffic accident understanding: (i) Negative Segment Referring (NSR), which utilizes contrastive learning to heighten the model's sensitivity to accident boundaries, and (ii) Video Spatial Alignment (VSA), which inserts spatial information into the training of video tasks, serving as a fine-grained complement to temporal localization. Followed by the functional tuning, we further generate multi-turn dialogues using an automated paradigm (Liu et al., 2024) and perform instruction tuning to enhance the dataset's utility and capabilities of MLLMs for human-like chatting and traffic accident understanding.

## 2 RELATED WORK

**Multimodal Large Language Models.** Extensive research has focused on enabling LLMs to process visual information, typically by adding an adapter between pre-trained visual models and LLMs to align features from different modalities (Li et al., 2023; Zhu et al., 2023; Liu et al., 2024). Some advanced multimodal LLMs, such as Qwen2-VL (Yang et al., 2024), unified image and video understanding into a single model, but still struggle with more fine-grained tasks. On the side of image

modal, object grounding has been a key focus, with a series of works (Chen et al., 2023b; Bai et al., 2023; Peng et al., 2023; Chen et al., 2023c; You et al., 2023) standardizing grounding coordinates to text format and achieving robust grounding capabilities. On the other hand, videos, as a more complex form of visual data, introduce greater challenges in aligning with video content (Maaz et al., 2023; Lin et al., 2023; Chen et al., 2023a; Zhang et al., 2023; Qian et al., 2024; He et al., 2024; Cheng et al., 2024; Xu et al., 2024; Zhang et al., 2024; Chen et al., 2023d; 2024). VTimeLLM (Huang et al., 2024) and TimeChat (Ren et al., 2024) address temporal localization by proposing time-aware attention mechanisms. GroundingGPT (Li et al., 2024) unified fine-grained capabilities across image and video for comprehensive multimodal understanding. Despite these advancements, previous works focus on general-purpose understanding, remaining largely unexplored in some specific scenarios, such as traffic accident understanding.

**Traffic Accident Detection and Understanding.** Traditional Traffic Accident Detection (TAD) methods are classified into single-stage (Hasan et al., 2016) and two-stage paradigms (Yao et al., 2019; Fang et al., 2022b). Single-stage approaches often rely on frame-to-frame errors but underperform in forecasting non-ego accidents and are sensitive to dynamic backgrounds (Hasan et al., 2016). Two-stage methods extract visual features, such as bounding boxes and optical flow, and apply TAD models to predict anomalies (Fang et al., 2022b). However, these methods depend heavily on the quality of feature extraction. Recent advances have integrated textual information into TAD, with TTHF (Liang et al., 2024) introducing text-driven attention mechanisms for anomaly detection in videos, and SUTD-TrafficQA (Xu et al., 2021) modeling fundamental question-answering and reasoning tasks for traffic accident scenes. On the MLLM front, empirical studies (Cao et al., 2023) have validated GPT-4(V)'s effective recall and description capabilities for traffic accident images. To this end, the potential of MLLMs in accident understanding remains unexplored, particularly in spatial-temporal grounding and reasoning over traffic accident videos.

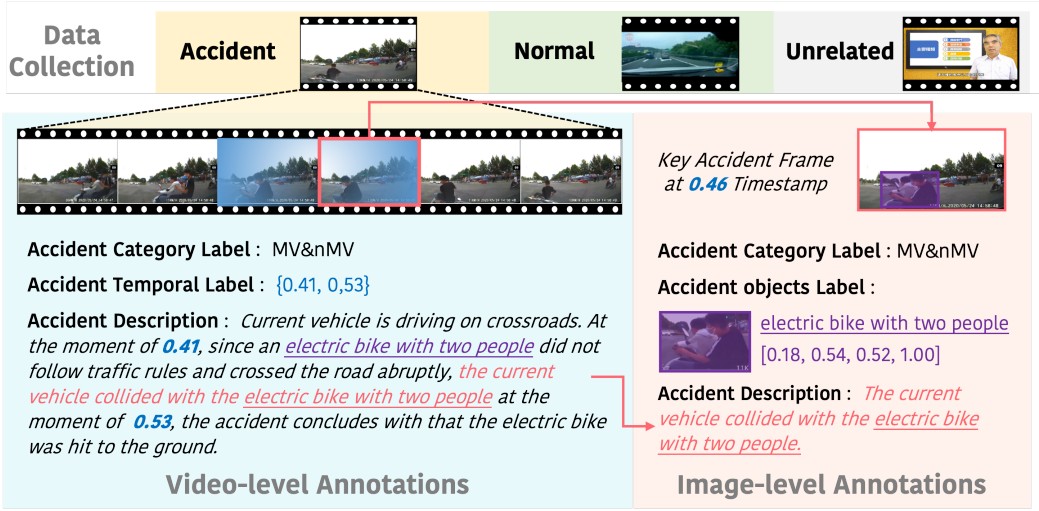

Figure 2: The data collection and annotation pipeline for building TAU-106K.

## 3 TAU-106K FOR VIDEO-IMAGE TRAFFIC ACCIDENT UNDERSTANDING

To advance the development of MLLMs for traffic accident analysis, we introduce TAU-106K, a comprehensive dataset integrating video and image data for traffic accident understanding, manually labeled with multi-granularity annotations through a video-to-image annotation pipeline (Figure 2).

### 3.1 VIDEO-BASED DATA COLLECTION AND ANNOTATION

**Video Data Collection and Preprocessing.** While traffic accident understanding is a critical public safety task and has been extensively studied, the available open-source benchmarks are limited in both scale and diversity, often featuring low-resolution video data. To address this, we aggregate established traffic accident benchmarks such as TAD (Xu et al., 2022), DoTA (Yao et al., 2022),

and CCD (Bao et al., 2020), selecting high-quality video clips as the data foundation for further annotation. We further expand the dataset by crawling road surveillance and dashcam footage from platforms like *YouTube* and *BiliBili*, capturing diverse real-world traffic conditions. Despite the abundance of traffic accident videos on the Internet, they are often unstructured and lack annotations. For the crawled raw videos, we first crop them into individual clips using the scene change detection toolkit `PySceneDetect`, and then manually filter out irrelevant or low-quality videos. Consequently, we obtain a collection of 51.5K traffic-focused video clips sourced from academic benchmarks and social media platforms, as illustrated in the upper part of Figure 2.

**Video-based Accident Annotations.** All existing benchmarks for traffic accident understanding lack comprehensive annotations, especially in terms of accident descriptions, which are crucial for enabling MLLMs to understand accident events in detail. To bridge this gap, we annotate from scratch or supplement existing annotations in three key aspects:

1. Accident Category: accident occurrence and detailed accident types. Each clip is reviewed to determine if an accident is present, labeled either as *Accident* or *Normal*. For *Accident* clips, we further categorize the accident type into five subcategories: single motor vehicle (SMV) accident, multiple motor vehicle (MMV) accident, multiple non-motor vehicle (MnMV) accident, motor vehicle and non-motor vehicle (MV&nMV) accident, and vehicle and pedestrian (V&P) accident.

2. Accident Duration: the specific time points of the accident occurrence. Annotators precisely identify the start and end timestamps of the accident within each clip, yielding the time points $\{t_{start}, t_{end}\}$. In particular, the start time $t_{start}$ should be the exact frame when the accident event begins, such as the moment of collision, while the end time $t_{end}$ is marked when the event concludes (e.g., stopping). Both timestamps are normalized within the clip duration to ensure consistency.

3. Accident Description: a detailed textual description of the nature of the accident, which is absent in all existing traffic accident benchmarks while being substantial for MLLMs to understand the accident event in detail. To ensure consistency and precision, we design a structured annotation template, guiding annotators to provide detailed and structured descriptions of the accident events.

The description template for *Accident* is structured to depict the **Traffic Scenario** (urban, highway, etc.), **Accident Content** including the objects involved in the accident (vehicles, pedestrians, etc.) and the nature of the accident (collision, scrape, etc.), and **Aftermath**, ensuring comprehensive and structured annotations. The labeled start and end timestamps are also incorporated into the description to provide temporal context for the accident event. Beyond the accident event itself, annotators are also encouraged to infer the **Potential Causes**, such as traffic rule violations or improper driving behaviors. The detailed template is also dependent on the **Footage Source**, either *Dashcam* or *Surveillance camera*. For intuitive understanding, we decompose the example shown in Figure 2 into the structured format in the gray block.

> [**Footage Source:** Current vehicle is driving on] [**Traffic Scenario:** crossroads]. At the moment of 0.41, [**Potential Causes:** since an electric bike with two people did not follow traffic rules and crossed the road abruptly], [**Accident Content:** the current vehicle collided with the electric bike with two people], at the moment of 0.53, [**Aftermath:** the accident concludes with that the electric bike was hit to the ground.]

In practice, these three annotation tasks are performed simultaneously, with multiple rounds of review and correction to ensure quality and consistency. This integrated approach ensures coherence in annotations, reflecting the interconnected nature of these tasks.

### 3.2 IMAGE-BASED DATA COLLECTION AND ANNOTATION

The above video annotations are multi-granularity while lacking spatial details, which are crucial for MLLMs to understand fine-grained visual features. To address this, we further derive images from the video clips and label them with detailed spatial information, whose detailed annotation pipeline is illustrated in the bottom right part of Figure 2.

**Image Data Collection and Selection.** Besides a few image-only accident datasets (e.g., Task-Fix (Juan et al., 2021)), most of the image data in our TAU-106K is sampled from the video clips. Guided by the temporal localization annotations in the video clips, we first extract candidate frames

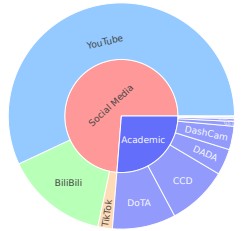 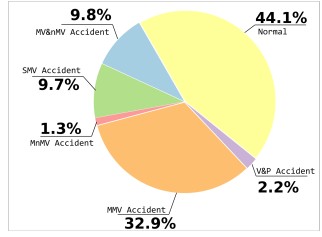 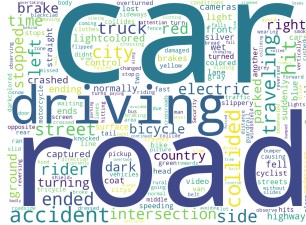

(a) Data source distribution.     (b) Accident type distribution.     (c) Word cloud of description.

Figure 3: Data source distribution, accident type distribution, and word cloud of accident descriptions in TAU-106K dataset.

by uniformly sampling frames within the labeled accident duration. These frames are then evaluated by annotators to select keyframes that best represent the accident events, based on the *Accident Description* in the video annotations. Notably, the time points of the selected keyframes are preserved to keep the temporal connection between the video and image data, which also enables our video spatial alignment strategy in model training. In addition to accident-related frames, we randomly sample accident-free frames to maintain data balance and prevent model bias.

**Image Annotations Derived from Video Annotations.** For the images sourced from existing benchmarks, we adopt the available annotations and extend them to our multi-granularity annotations. On the other hand, for the images derived from video data, we inherit the accident-related annotations from the video clips, including the *Accident Category* and *Accident Description*, where the latter extracted from the *content of the accident* part in the video-based accident description to maintain annotation consistency and reduce workload. In particular, labels for the involved objects are derived directly from the accident descriptions, ensuring that the annotated objects are those explicitly mentioned. For instance, given the accident description as "*A blue car collides with a pedestrian in white clothes*", the corresponding objects will be labeled as *blue car* and *pedestrian in white clothes*, respectively. This instance-specific labeling helps MLLMs focus on the objects directly involved in the accident, minimizing distractions from irrelevant objects of the same category that may appear in the scene.

Table 1: The comparison of TAU-106K with other accident-specific or general-purpose benchmarks. 'CLS': Accident Categories, 'TL': Temporal Annotation, 'Bbox': Object Grounding Annotation, 'CAP': Caption Annotation, and 'QA': Question-Answer Pairs.

| Dataset | Years | Domain | # Videos | Annotations | Avg. Words | Avg. Duration |
|---|---|---|---|---|---|---|
| Dashcam (Chan et al., 2017) | 2016 | Traffic | 3,000 | TL; | - | 5.0 seconds |
| A3D (Yao et al., 2019) | 2019 | Traffic | 1,500 | TL; | - | 8.5 seconds |
| CCD (Bao et al., 2020) | 2021 | Traffic | 1,500 | TL; | - | 5.0 seconds |
| TAD (Lv et al., 2021) | 2021 | Traffic | 500 | CAP; TL; | - | 35.8 seconds |
| DADA (Fang et al., 2021) | 2021 | Traffic | 200 | CAP; TL; Driver Attention | - | 11.0 seconds |
| SUTD-TrafficQA (Xu et al., 2021) | 2021 | Traffic | 10,080 | QA pairs | - | 13.6 seconds |
| DoTA (Yao et al., 2022) | 2022 | Traffic | 4,677 | CAP; TL; Bbox | - | 15.6 seconds |
| CAP (Fang et al., 2022a) | 2023 | Traffic | 11,727 | CAP; TL; Fixed-Form CAP | 6.3 | 6.2 seconds |
| **TAD-106K (Ours)** | 2024 | Traffic | 51,544 | CAP; TL; Bbox; Free-Form CAP | 32.1 | 10.3 seconds |
| Charades-STA (Gao et al., 2017) | 2017 | Daily | 9,848 | TL; Free-Form CAP | 6.3 | 31 seconds |
| DiDeMo (Anne Hendricks et al., 2017) | 2017 | Open | 10,464 | TL; Free-Form CAP | 7.5 | 30 seconds |
| ActivityNet-Captions (Krishna et al., 2017) | 2017 | Open | 19,209 | TL; Free-Form CAP | 13.5 | 180 seconds |

## 3.3 DATA STATISTICS AND ANALYSIS

TAU-106K comprises 106K multimodal data instances, including 51.5K video clips and 54.8K images, all with high-quality annotations. The majority of the video clips and images are in 720p resolution and are sourced from both open-source benchmarks and social media platforms, as shown in Figure 3(a). Among the TAU-106K, 56% of instances are labeled as *Accident* and 44% as *Normal*, with detailed category distribution shown in Figure 3(b). The balanced distribution of accident-related and accident-free instances ensures that the model is trained robustly, avoiding biases towards accident occurrences. The average video duration of processed and filtered clips is 10.3 seconds, with annotated accidents lasting an average of 3 seconds (approximately 25% of the video clip). As for the image data, 45K accident-involved objects are grounded, with an average of 1.6 bounding boxes per image and an average bounding box area covering 7.9% of the image. Our accident de-

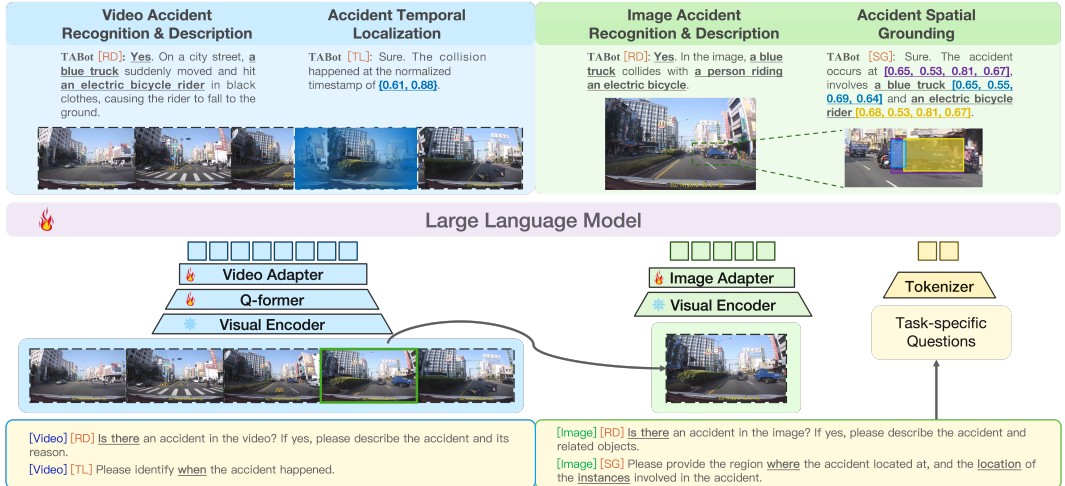

Figure 4: The model architecture and functional capabilities of the TABot.

scriptions are detailed and diverse, covering a broad range of traffic scenarios, accident types, and objects involved, as shown in the word cloud of accident descriptions in Figure 3(c).

We provide a more comprehensive comparison between TAU-106K and other datasets, focusing on key features such as size, domain, annotation types, and the characteristics of the textual captions, depicted in Table 1. According to the comparison, our TAU-106K is the largest dataset in terms of the number of videos and the diversity of annotations, supporting a wide range of tasks. In particular, benefiting from our manual annotation process that is labor-intensive yet worthy, the labeled free-form accident captions in TAU-106K are much more diverse and detailed than other datasets, achieving a largest average length of 32.1 words per caption. This makes TAU-106K a valuable resource for training and evaluating accident-aware models in traffic video understanding.

## 4    TABOT: A CHATBOT FOR TRAFFIC ACCIDENT UNDERSTANDING

We introduce TABot, a multimodal fine-grained MLLM developed by leveraging instructional data constructed from the TAU-106K dataset. TABot is compatible with both video and image modalities, enabling it to perform fine-grained understanding and reasoning tasks in traffic accident scenarios. The proposed TABot integrates a suite of traffic accident-related tasks, as depicted in Figure 4.

### 4.1    MODEL OVERVIEW

We advance the TABot upon GroundingGPT (Li et al., 2024), a model known for its strong performance in fine-grained image and video understanding. By fine-tuning this general-purpose MLLM on our annotated TAU-106K dataset, we enhance its capabilities for traffic accident comprehension on several functional tasks including Accident Recognition, Accident Description Generation, Accident Temporal Localization, and Accident Spatial Grounding, as illustrated in Figure 4. It is notable that we normalize the responses to the temporal localization and spatial grounding tasks to the video duration and image size, respectively, to ensure consistency and facilitate model training. The normalized responses are denoted as $\{t_{start}, t_{end}\}$ for temporal localization and $[x_{min}, y_{min}, x_{max}, y_{max}]$ for spatial grounding, enclosed in specific tokens to indicate the temporal boundaries and spatial regions.

Following previous works (Li et al., 2024), we adopt a two-stage fine-tuning approach: Firstly, during the functional tuning stage, TABot is jointly fine-tuned on both image and video data, focusing on the four key tasks mentioned above. We generate structured single-round conversations for each task to facilitate the model's understanding of traffic accidents from different perspectives. Two additional training strategies are proposed to further improve performance in temporal localization: Negative Segment Referring (NSR) and Video Spatial Alignment (VSA), which promote the performance from the perspective of contrastive learning and spatial understanding, respectively. NSR samples accident-free segments before the occurrence of an accident and trains the model by referring to the sample segments containing no accidents, serving as negative data to highlight the

perception of accident occurrences. On the other hand, benefiting from the unified video-to-image annotation pipeline, VSA involves the spatial grounding annotations into the video training process, complementing spatial information from images into the temporal localization task. As for the implementation details, we extend the answer to the temporal localization task to include the spatial grounding annotations. For example, the response to the temporal localization task '{*0.30, 0.45*}' may be further extended with "*At the timestamp 0.38, an accident occurs at the region of [0.21, 0.35, 0.87, 0.57].*" This alignment improves TABot's fine-grained spatial understanding of accidents in video contexts. Additionally, to ensure the model's flexibility in handling multiple tasks, task-specific flag tokens (Accident Recognition & Description [RD], Temporal Localization [TL], and Spatial Grounding [SG]) are inserted at the start of each query to guide TABot's responses.

With the above functional tuning, TABot is endowed with the capabilities to perform coarse- and fine-grained traffic accident understanding tasks. To further advance the TABot's comprehensive understanding and conversational skills, we utilize the textual captions of the video clips and images as the abstracts to prompt the powerful LLMs (Achiam et al., 2023), to conclude the above functional tasks and generate additional accident-oriented dialogue, such as the causes of accidents or prevention suggestions. The generated multi-round dialogue set is then used to instruct the TABot model. Through such instruction tuning, TABot is upgraded to a chat version (TABot-Chat) with enhanced instruction-following capabilities and a more comprehensive understanding of traffic accidents. In particular, the task flag tokens are maintained in the instruction tuning stage to guide the model's responses to specific tasks, ensuring the functional capabilities of the model towards multiple accident-oriented tasks.

## 5 EXPERIMENTS

We set GroundingGPT-7B (Li et al., 2024), a pre-trained general-purpose MLLM with temporal and spatial grounding capabilities, as the baseline model for our TABot. The detailed experimental settings of the two-step approach are described as follows:

**Functional Tuning.** We train LLM and both visual adapters of the GroundingGPT model through our TAU-106K dataset for 3 epochs using $8 \times$ H800 GPUs. The initial learning rate is set to 2e-5 with a batch size of 32, requiring about 20 hours to complete.

**Instruction Tuning.** We extend training with the instruction-tuning dataset generated by LLaMA-70B (Dubey et al., 2024), leading to our TABot-Chat model. To avoid catastrophic forgetting, we combine the single-round and multi-round dialogue conversations during this stage, further training the model for 1 epoch on $8 \times$ H800 GPUs for about 9 hours with the learning rate and batch size unchanged.

**Evaluation Metrics.** For evaluation purposes, the TAU-106K dataset was split into training and testing sets in a 9:1 ratio, ensuring the same distribution of normal/accident instances and scene continuity across both. We evaluate the TABot on four functional tasks for both image and video data. The evaluation metrics are as follows:

*1) Accident Recognition.* Recall, Precision, and F1 scores are used to assess the model's accuracy in distinguishing accidents from normal scenes in both image-level and video-level contexts.

*2) Accident Description.* BLEU-1 score, Rouge-L F1 score, and BERT F1 score are employed to measure the model's ability to generate coherent and accurate accident descriptions. We further leverage GPT-4o to estimate the quality of the generated descriptions, referred to as GPT-4 score.

*3) Accident Temporal Localization.* We reported the Intersection over Union (IoU) between predicted and true temporal intervals, along with Average Precision (AP@30, AP@50, AP@70).

*4) Accident Spatial Grounding.* We evaluate the model's performance on accident region and object grounding through reporting detection metrics: mean Intersection over Union (mIoU) and Average Precision (AP@30, AP@50, AP@70).

### 5.1 VIDEO-LEVEL TASKS

In this subsection, we present the results on video-level tasks of our proposed models, including TABot, TABot-Chat, and their comparison with several existing methods: Video-LLaVA (Lin et al.,

2023), TimeChat (Ren et al., 2024), VTimeLLM (Huang et al., 2024), GroundingGPT (Li et al., 2024), Qwen2-VL (Wang et al., 2024), and Gemini-1.5-Pro (Reid et al., 2024).

Table 2: Experimental results on video accident recognition in traffic scenes. "@A" and "@N" represent the class-wise results on accidents and normal scenes.

| Methods | Video Accident Recognition | | | | | | |
|---|---|---|---|---|---|---|---|
| | Acc | Rec@A | Pre@A | F@A | Rec@N | Pre@N | F1@N |
| Video-LLaVA (Lin et al., 2023) | 50.20 | 99.70 | 50.10 | 66.69 | 0.70 | 70.00 | 1.39 |
| TimeChat (Ren et al., 2024) | 54.65 | 91.80 | 52.67 | 66.93 | 17.50 | 68.09 | 27.84 |
| VTimeLLM (Huang et al., 2024) | 50.00 | **100.00** | 50.00 | 66.67 | 0.00 | 0.00 | 0.00 |
| GroundingGPT (Li et al., 2024) | 50.00 | **100.00** | 50.00 | 66.67 | 0.00 | 0.00 | 0.00 |
| Qwen2-VL (Wang et al., 2024) | 72.65 | 53.46 | 87.23 | 66.29 | 92.08 | 66.16 | 77.00 |
| Gemini-1.5-Pro (Reid et al., 2024) | 69.61 | 61.82 | 74.18 | 67.44 | 77.70 | 66.25 | 71.52 |
| **TABot (Ours)** | 81.00 | 78.65 | 85.10 | 81.75 | 83.77 | 76.90 | 80.19 |
| **TABot-Chat (Ours)** | **82.05** | 79.70 | 86.00 | **82.73** | 84.80 | **78.10** | **81.31** |

As presented in Table 2, most previous models struggle to recognize traffic accidents, with accuracies ranging from 50% to 54.65%. VTimeLLM and our baseline Grounding tend to classify all videos as abnormal, indicating several false positives. Although Qwen2-VL and Gemini-1.5-Pro show some improvement, they tend to classify the videos as normal, exhibiting a lack of accident perception. In contrast, our TABot, trained on our TAU-106K dataset, demonstrates a significant improvement, reaching an accuracy of 80.95% and outperforming all prior methods. Further instruction tuning with multi-round dialogue data, our TABot-Chat variant further presents an accuracy of 82.05% and improved overall performance for both accident and normal scenarios.

Table 3: Experimental results on video accident description and accident temporal localization.

| Methods | Video Accident Description | | | | Accident Temporal Localization | | | |
|---|---|---|---|---|---|---|---|---|
| | BLEU | Rouge | BERT | GPT-4 | AP@30 | AP@50 | AP@70 | mIoU |
| Video-LLaVA (Lin et al., 2023) | 22.20 | 24.81 | 60.72 | 26.17 | - | - | - | - |
| TimeChat (Ren et al., 2024) | 7.12 | 18.16 | 58.77 | 12.67 | 23.00 | 7.90 | 2.50 | 18.07 |
| VTimeLLM (Huang et al., 2024) | 25.25 | 23.32 | 60.84 | 18.62 | - | - | - | - |
| GroundingGPT (Li et al., 2024) | 9.77 | 16.43 | 55.70 | 14.00 | 4.60 | 2.40 | 0.90 | 3.79 |
| Qwen2-VL (Wang et al., 2024) | 15.38 | 23.64 | 61.61 | 39.80 | 32.91 | 15.76 | 5.42 | 20.75 |
| Gemini-1.5-Pro (Reid et al., 2024) | 12.83 | 19.57 | 60.79 | 23.66 | 13.87 | 5.14 | 1.64 | 9.31 |
| **TABot (Ours)** | 54.59 | 57.94 | 82.31 | 55.60 | **39.44** | 20.12 | **9.80** | **25.93** |
| **TABot-Chat (Ours)** | **55.70** | **58.32** | **83.78** | **55.73** | 37.90 | **20.70** | 7.80 | 25.33 |

For the tasks of video accident description and temporal localization, the performance of our models is detailed in Table 3. TABot excels in generating accurate and contextually relevant accident descriptions, achieving the highest BERT and GPT-4 scores, indicating high semantic alignment with human judgments. As for the most challenging task of accident temporal localization, previous models struggled to pinpoint the occurrence of accidents, and only Qwen2-VL demonstrated a certain capability in fine-grained localization within videos. Our TABot also significantly surpasses all existing methods in this fine-grained task, establishing a new state-of-the-art in temporal localization performance. After instruction tuning, the TABot-Chat variant shows improved description capabilities as we expected, with a slight decrease in temporal localization performance. This decrease can be attributed to the model's enhanced conversational abilities, which may lead to a slight decline in the model's focus on temporal localization.

## 5.2 IMAGE-LEVEL TASKS

In addition to the video-level tasks, we also evaluate our proposed models on image-level tasks. The experimental results are presented in Tables 4 and 5, where we compare our models against several state-of-the-art methods (Zhu et al., 2023; Li et al., 2024; Bai et al., 2023; Wang et al., 2024; Reid et al., 2024; Achiam et al., 2023).

Table 4 presents the results of the image accident recognition. Our TABot outperforms all methods in image accident recognition, demonstrating the quality of our dataset and the effectiveness of our training strategies. As for accident description, the superior performance of our models is evident, validating that our model excels in generating accurate and contextually relevant descriptions of accidents. TABot-Chat, following instruction tuning, attains excellent values of 77.26 and 55.73 for BERT and GPT-4 scores, indicating high semantic alignment with human judgments.

Table 4: Experimental results on image accident recognition and description in traffic scenes.

| Methods | Image Accident Recognition | | | | | | | Image Accident Description | | | |
|---|---|---|---|---|---|---|---|---|---|---|---|
| | Acc | Rec@A | Pre@A | F1@A | Rec@N | Pre@N | F1@N | BLEU | Rouge | BERT | GPT-4 |
| MiniGPT4 (Zhu et al., 2023) | 64.05 | 75.57 | 68.89 | 72.08 | 45.73 | 54.06 | 49.54 | 9.63 | 11.56 | 45.84 | 11.67 |
| GroundingGPT (Li et al., 2024) | 63.75 | 79.15 | 67.45 | 72.84 | 39.25 | 54.20 | 45.53 | 7.22 | 7.81 | 45.00 | 21.08 |
| Qwen-VL-Max (Bai et al., 2023) | 69.95 | 87.87 | 70.48 | 78.22 | 41.45 | 68.23 | 51.57 | 4.59 | 4.27 | 43.08 | 28.46 |
| Qwen2-VL (Wang et al., 2024) | 58.35 | 40.07 | 83.53 | 54.16 | 87.44 | 47.84 | 61.84 | 23.31 | 24.53 | 66.12 | 32.01 |
| Gemini-1.5-Pro (Reid et al., 2024) | 80.99 | 0.00 | 0.00 | 0.00 | 80.99 | 1.00 | 89.50 | 16.28 | 21.53 | 64.44 | 24.54 |
| GPT-4o (Achiam et al., 2023) | 63.65 | 45.44 | 90.73 | 60.55 | 92.62 | 51.62 | 66.30 | 4.78 | 5.18 | 43.05 | 35.71 |
| **TABot (Ours)** | **90.75** | 94.38 | **90.31** | **92.30** | 85.58 | 91.45 | **88.42** | 48.62 | 43.31 | 75.20 | 55.12 |
| **TABot-Chat (Ours)** | 90.50 | **94.90** | 89.33 | 92.03 | 84.48 | **92.36** | 88.24 | **50.28** | **45.67** | **77.26** | **55.73** |

Table 5: Experimental results on accident region and object grounding in traffic images.

| Methods | Accident Region Grounding | | | | Accident Object Grounding | | | |
|---|---|---|---|---|---|---|---|---|
| | AP@30 | AP@50 | AP@70 | mIoU | AP@30 | AP@50 | AP@70 | mIoU |
| MiniGPT4 (Zhu et al., 2023) | 50.57 | 34.85 | 24.67 | 39.36 | 70.33 | 56.65 | 33.24 | 49.72 |
| GroundingGPT (Li et al., 2024) | 26.55 | 14.25 | 7.82 | 3.84 | 62.23 | 49.06 | 27.34 | 43.75 |
| Qwen-VL-Max (Bai et al., 2023) | 43.73 | 26.47 | 12.79 | 30.72 | 59.97 | 45.27 | 28.25 | 43.00 |
| Qwen2-VL (Wang et al., 2024) | 60.21 | 47.52 | 29.70 | 43.02 | 71.66 | 57.48 | 35.66 | 50.38 |
| Gemini-1.5-Pro (Reid et al., 2024) | 56.66 | 37.20 | 17.42 | 37.85 | 46.07 | 34.99 | 20.09 | 31.98 |
| **TABot (Ours)** | 80.05 | **70.03** | **45.52** | **57.83** | **78.05** | **65.86** | **39.88** | **54.95** |
| **TABot-Chat (Ours)** | **80.29** | 69.87 | 44.95 | 57.63 | 77.64 | 65.41 | 39.68 | 54.78 |

Table 5 showcases the results for region- and object-level grounding. Our TABot significantly outperforms the baselines in terms of AP and mIoU for both accident regions and objects. Similar to the phenomenon observed in video tasks, instruction tuning slightly perturbs the model's grounding performance, but the overall performance remains competitive. The essential fine-grained grounding performance gap compared to previous methods further highlights the necessity of collecting traffic accident data and training models on this specific domain.

## 5.3 ABLATION STUDY

**The Effectiveness of Joint Training.** To evaluate the impact of joint training on image and video data, we additionally train the TABot using a single modality (TABot-single trained on image or video data only) and compare the results with our joint training model (TABot). According to the main results in Tables 6, the joint training model outperforms the single modality models in most tasks, especially for image-level tasks. The improvement in accident recognition and description tasks is more pronounced than in spatial grounding tasks, indicating that the primary benefit of video data is the scale-up in the amount of training data, which is particularly effective for tasks requiring richer contextual information. On the other hand, incorporating image data into video tasks leads to a minor performance drop, suggesting that the model's focus on video data may have slightly compromised its performance on image tasks.

**The Effectiveness of VSA and NSR.** Benefiting from our unified video-image annotation pipeline, VSA can explicitly incorporate spatial grounding annotations at specific time frames into the training of video temporal localization. As shown in Table 7, our VSA strategy leads to a consistent improvement in the model's temporal localization capabilities, demonstrating its effectiveness in involving spatial information as a complementary signal to enhance the model's temporal perception. As for NSR, it improves the model's overall performance across both image and video tasks by enhancing its capacity to differentiate accident events from normal content, as indicated in Table 7. However, there is a marginal decline in spatial grounding performance, and we attribute this to the model's focus on temporal localization, which may have led to a slight trade-off in spatial understanding. This drawback is compensated when the NSR is combined with the VSA, as all tasks achieve their best performance, demonstrating the complementary nature of these two strategies.

Table 6: Ablation study of separate (TABot-single) or joint (TABot) training on image and video data. "AG", "OG" & "TL" denote the AP@50 of Accident region Grounding, accident Object Grounding, and Temporal Localization.

| Model | Image Understanding | | | | | Video Understanding | | | |
|---|---|---|---|---|---|---|---|---|---|
| | Acc | BERT | GPT-4 | AG | OG | Acc | BERT | GPT-4 | TL |
| TABot-single | 77.95 | 74.16 | 48.22 | 68.97 | 64.70 | 80.95 | **82.62** | 54.63 | **20.28** |
| TABot | **90.75** | **75.20** | **55.12** | **70.03** | **65.86** | **81.00** | 82.31 | **55.60** | 20.12 |

Table 7: Ablation study on the additional training strategies.

| TABot | | Image Understanding | | | | | Video Understanding | | | |
|---|---|---|---|---|---|---|---|---|---|---|
| VSA | NSR | Acc | BERT | GPT-4 | AG | OG | Acc | BERT | GPT-4 | TL |
| ✗ | ✗ | 88.45 | 75.09 | 54.28 | 68.57 | 64.06 | 80.50 | 82.08 | 55.23 | 19.30 |
| ✗ | ✓ | 88.00 | 74.73 | 53.82 | 70.20 | 64.21 | **81.90** | 81.72 | 54.78 | 18.90 |
| ✓ | ✗ | 88.60 | 74.83 | 53.91 | **70.36** | 64.55 | 80.80 | 82.26 | 55.53 | 19.92 |
| ✓ | ✓ | **90.75** | **75.20** | **55.12** | 70.03 | **65.86** | 81.00 | **82.31** | **55.60** | **20.12** |

Table 8: Ablation study on the training strategy of the **instructing tuning**.

| TABot-Chat | | Image Understanding | | | | | Video Understanding | | | |
|---|---|---|---|---|---|---|---|---|---|---|
| Mixed Data | Task Flag | Acc | BERT | GPT-4 | AG | OG | Acc | BERT | GPT-4 | TL |
| ✗ | ✗ | 84.55 | 75.44 | 50.18 | 68.71 | 64.52 | 79.50 | 82.43 | 53.32 | 5.10 |
| ✗ | ✓ | 85.50 | 75.59 | 52.83 | 69.14 | 64.76 | 79.35 | 82.14 | 54.51 | 13.30 |
| ✓ | ✗ | 88.30 | 76.56 | 52.04 | 69.22 | 64.11 | 80.20 | 83.10 | 55.40 | 18.90 |
| ✓ | ✓ | **90.45** | **77.20** | **55.73** | **69.46** | **64.96** | **81.25** | **83.51** | **55.73** | **19.50** |

**Training Strategies for Chat Version.** In the TABot-Chat model, we observe that directly performing instruction tuning without additional recipes significantly degrades the model's performance in functional tasks, for example, the accuracy for image accident recognition decreased to 84.55%. To maintain or even improve the functional performance, we took some data-centric approaches: (1) mix the datasets used for Functional Tuning and Instruction Tuning. (2) introduce task flags to specify the target response for the model in a multi-task framework. As presented in Table 8, based on our training data paradigm, we successfully improve the conversational performance of TABot-Chat while maintaining excellent functional results.

Table 9: The ablation study of reasoning captions on the temporal localization task.

| Model | AP@30 | AP@50 | AP@70 | mIoU |
|---|---|---|---|---|
| TABot | 39.44 | 20.12 | 9.80 | 25.93 |
| - Reason Caption | 34.20 | 16.90 | 6.60 | 21.67 |

**Effectiveness of Reasoning Description.** In the application of MLLMs to traffic accident understanding, the most critical task is to achieve precise temporal localization of accidents in videos. The labeled reason caption in our TAU-106K dataset is a portent of the content of the accident, making accident detection and localization more trackable. Here we evaluate the effectiveness of the reasoning caption in the temporal localization task by conducting an ablation study as shown in Table 9. The results show that the removal of reasoning captions leads to a significant performance drop in the temporal localization task, validating our claim that reasoning captions serve as valuable cues for accident understanding. Our future work will focus on developing more reasoning tasks based on the reasoning captions in TAU-106K to achieve accident forecasting and causality analysis tasks.

## 6 DISCUSSION AND CONCLUSION

To advance the exploration of MLLMs for traffic accident understanding, we introduced video-image-text joint dataset TAU-106K, which includes 51.5K video clips and 54.8K images, with high-quality annotations covering coarse- and fine-grained accident-oriented information. Upon our comprehensive dataset, we proposed TABot, a unified MLLM that is compatible with video and image data and can handle various traffic accident understanding tasks including accident recognition, description, temporal localization, and spatial grounding. Our method and dataset lay the foundation for MLLM to infer and understand fine-grained representations of traffic accident scenarios. Our publicly available data and code will facilitate further research on MLLM for traffic accidents. Future work will include more detailed grounding and addressing the hallucination problem.

ACKNOWLEDGMENTS

This work was partially supported by the National Natural Science Foundation of China under Grants No. 62476201 and the New Cornerstone Science Foundation through the XPLORER PRIZE.

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

## A  ADDITIONAL DETAILS ON DATA COLLECTION AND STATISTICS

### A.1  DATA COLLECTION AND ANNOTATION

**Data Source.** We collect 106K media data, including 51,544 video clips and 54,767 images sourced directly or extracted from the video clips. In detail, as shown in Table 10, our dataset comprises 13,536 video clips and 34,968 images from 9 open-source traffic benchmarks, and 38,008 video clips and 19,799 images from social media platforms.

Table 10: The detailed distribution of our TAU-106K dataset from different data sources.

| Data Source | # Video | # Image |
|---|---|---|
| DoTA (Yao et al., 2022) | 4,672 | 9,502 |
| CCD (Bao et al., 2020) | 4,464 | 6,010 |
| DADA (Fang et al., 2021) | 1,923 | 3,799 |
| DashCam (Chan et al., 2017) | 1,727 | 2,348 |
| TAD-1 (Lv et al., 2021) | 352 | 628 |
| TAD-benchmark (Xu et al., 2022) | 208 | 546 |
| Drive-Anomaly106 (Zhu et al., 2019) | 105 | 207 |
| RetroTrucks (Haresh et al., 2020) | 56 | 348 |
| TrafficS (Ghahremannezhad et al., 2022) | 29 | 58 |
| SUTD-TrafficQA (Xu et al., 2021) | — | 9,674 |
| CADP (Shah et al., 2018) | — | 914 |
| TaskFix (Juan et al., 2021) | — | 713 |
| YouTubeCrash (Kim et al., 2019) | — | 221 |
| *Youtube* (Social Media Platform) | 29,364 | 12,345 |
| *BiliBili* (Social Media Platform) | 7,577 | 6,476 |
| *TikTok* (Social Media Platform) | 1,067 | 978 |

**Data Annotation Process.** We collaborate with a professional data annotation team to conduct the annotation of all traffic video and image data. The overall annotation process is divided into two parts: video-based annotation and image-based annotation. Each part is separated into annotation and verification phases carried out by different annotators. The annotators in the verification phases are tasked with verifying the quality of each data item as "satisfactory" or "unsatisfactory". Unsatisfactory items were sent back to the annotation pipeline for refinement.

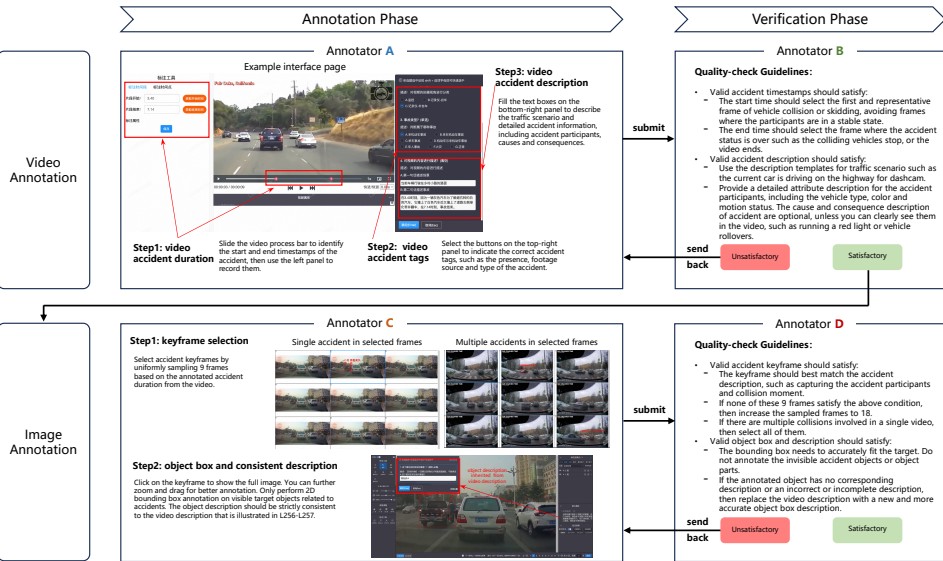

Figure 5: The annotation process and quality control of the TAU-106K dataset.

We use an internal annotation tool to enable interactive use with our annotators and a diagram of the annotation protocol used in our data engine, which is illustrated in Figure 5. Specifically, annotator A in the video annotation phase adopts Steps 1, 2 and 3 to provide the timestamps, semantic tags and detailed description for accident video, and another annotator B focuses on quality verification.

After the video annotation, the fine video data is sent to the image annotation phase, annotator C utilizes Steps 1 and 2 to perform keyframe selection and accident-related object annotation, and then annotator D conducts the image-level quality verification. We employed a team of 50 experienced annotators and all of them followed the same annotation guidelines presented in both video and image verification phases. According to our annotation workflow, each data item involved at least four different annotators to uphold a high standard for annotation. Moreover, benefiting from our proposed video-to-image annotation pipeline, the image annotators and verifiers double-check the annotations from the video phase, ensuring the consistency and accuracy of the annotations across different modalities, which is also label-efficient and cost-effective. We present some annotation samples in Figure 6.

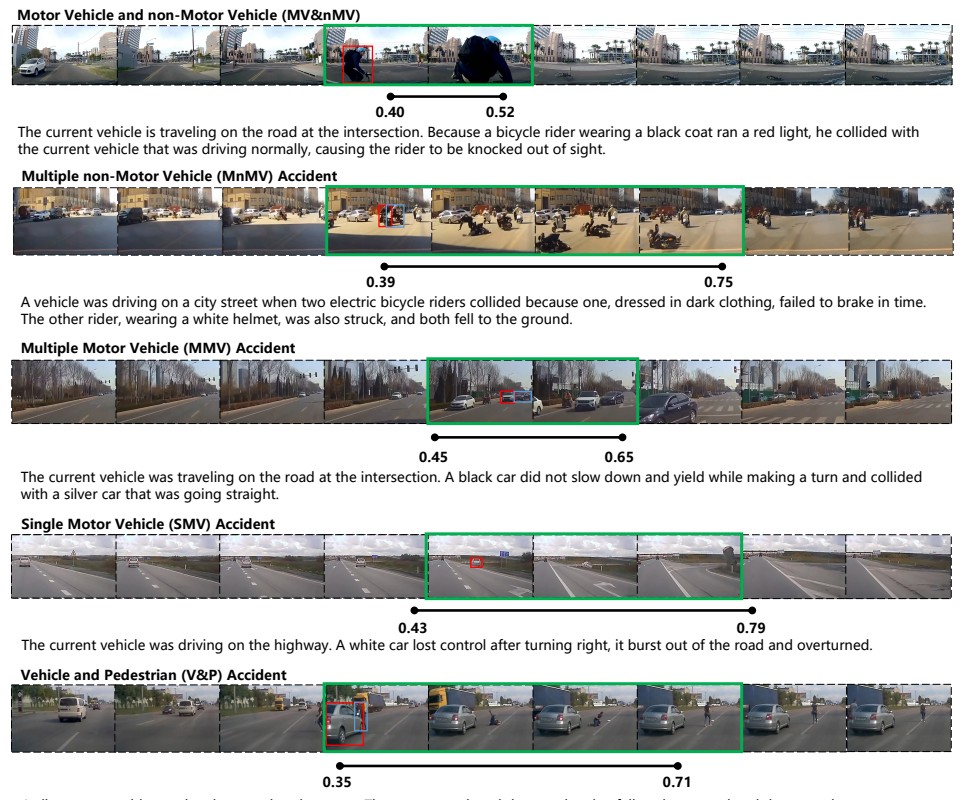

Figure 6: The video annotation examples of different types of accidents.

## A.2 DATA STATISTICS

**Data Categories.** Our dataset covers various categories of traffic scenarios, objects, and accidents:

- *Traffic Scenarios*: urban streets (49%), intersections (19%), country roads (17%), highways (12%), and other traffic scenes (3%).
- *Objects:* cars (58%), trucks (12%), electric bikes (11%), pedestrians (5%), vans (3%), bicycles (3%), buses (2%), guardrails (2%), motorcycles (2%), and other objects (2%).
- *Accident Categories*: multi-motor-vehicle accidents (59%), motor-vehicle & non-motor-vehicle accidents (18%), single-motor-vehicle accidents (17%), vehicle & pedestrian accidents (4%), and multi-non-motor-vehicle accidents (2%).

**Video Duration Distribution.** As shown in Figure 7 (a), the video duration distribution of the accident-related video clips is visualized. The video clips collected from previous benchmarks or cropped from the raw crawled videos are relatively short, with the majority of the video clips lasting less than 20 seconds. Rarely, some video clips exceed 50 seconds, with the longest video clip lasting 12 minutes. For better visual presentation, we restrict the x-axis to 50 seconds, which covers the majority of the video clips.

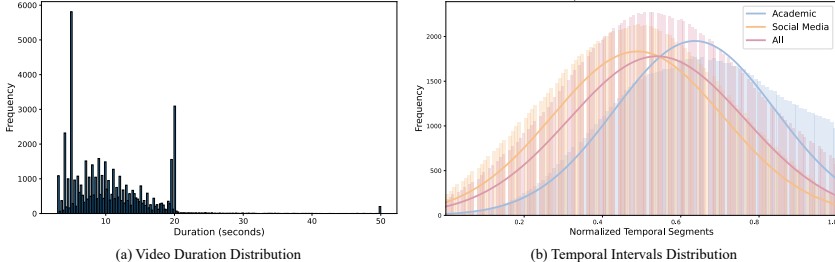

(a) Video Duration Distribution  (b) Temporal Intervals Distribution

Figure 7: (a) Video duration distribution of the accident-related video clips; (b) Temporal intervals distribution of the accident events in the video clips. "Frequency" refers to the number of occurrences of videos with a certain length in the TAU-106K dataset.

**Temporal Localization Distribution.** To statistically examine the temporal distribution of accident events in the video clips, we analyze and visualize the annotated temporal intervals of these events, as illustrated in Figure 7 (b). The temporal intervals corresponding to accident events in existing benchmarks tend to exhibit a relatively concentrated distribution, with the majority of accidents occurring towards the later portions of the video clips. In contrast, the temporal intervals of accident events in the newly collected video clips display a more dispersed pattern, resembling a Gaussian-like distribution. This broader temporal spread results in a more balanced distribution of accident intervals across the video clips. Such a distribution helps to mitigate potential biases toward specific temporal segments, enabling MLLMs to more effectively and authentically learn the temporal dynamics and characteristics of accident events. By ensuring a diverse temporal representation, the proposed dataset enhances the robustness and generalizability of temporal localization models.

# B ADDITIONAL DETAILS ON MODEL TUNING DATA

## B.1 FUNCTIONAL TUNING DATA TEMPLATES

To avoid the trained model only responding to the specific instruction in the training data, we predefined a set of questions for each task to facilitate the model to be activated facing diverse queries.

**Single-turn: Temporal Localization.** Besides the diverse question set, we provide a set of answer templates to prompt the model to generate human-like responses in the temporal localization task:

```python
question_templates = [
    "Do you know the exact times the traffic accident kicked off and wrapped up?",
    "Can you give me the start and end times of the traffic accident in the video?",
    "Any idea about the start and end time of that traffic accident we saw?",
    "Show me when the traffic accident gets going and when it's all over?",
    "What is the start and end time of the traffic accident in the video?",
    "Could you specify the timing of traffic accident's onset and conclusion?",
    "Please specify the precise timing of the traffic accident's onset and conclusion.",
    "At what timestamps does the traffic accident commence and finish?",
    "Can you delineate the duration of the traffic accident from beginning to end?",
    "When is the traffic accident initiated and terminated in the footage?"
]

answer_templates = [
    "Between {}.".format,
    "In the time period {}.".format,
    "During the span of {}.".format,
    "It happens in {}.".format,
    "At {}.".format,
    "Exactly at {}.".format,
    "Through {}.".format,
    "Within the window of {}.".format,
    "In the {} mark.".format,
    "Around {}.".format
]

User: random.choice(question_templates) <video>

GPT: random.choice(answer_templates)(annotation["accident-segments"])
```

Additionally, as we introduced in the main text, we also prompt the model to describe the content within the particular temporal segments: random-sampled normal segments or labeled accident segments. The pseudo code of temporal referring question-answer pair generation is presented:

```
question_templates = [
    "What's happened during {} in the video?".format,
    "What's the incident in the period of {}?".format,
    "Maybe something wrong happened during {} in the provided video?".format,
    "What's the traffic situation in the period of {}?".format,
    "Is the traffic flow captured by the video normal during {}?".format,
    "Dose the accident happen during {} in the video?".format,
    "Does the video record any traffic disruptions or accidents around {}?".format,
    "Is there any indication of an abnormal traffic event during {}?".format,
    "Could you identify any mishaps in the time frame of {}?".format,
    "Are there signs of vehicular distress or accidents within {}?".format
]

User: random.choice(question_templates)(annotation["accident_segments"]) <video>

GPT: annotation['accident_description']
```

**Single-turn: Accident Spatial Grounding.** There are two spatial grounding tasks during our training process: accident-involved object grounding and accident region grounding. As for the accident-involved object grounding task, the pseudo code of generating conversations is presented as follows:

```
question_templates = [
    "Where is the {} involved in the accident?".format,
    "Where is the {} involved in the accident in the image?".format,
    "Provide the coordinates of the {} involved in the accident in the image?".format,
    "Can you point out the {} involved in the accident in the image and provide the coordinates
        of its location?".format,
    "Help me to locate the {} involved in the accident in the image and give me its coordinates
        , please.".format,
    "In the given image, could you find and tell me the coordinates of the {} involved in the
        accident?".format,
    "Guide me to the location of the {} involved in the accident within the image by providing
        its coordinates.".format,
    "I'd like to know the exact coordinates of the {} involved in the accident in the photo.".
        format,
    "Would you kindly provide the coordinates of the {} involved in the accident located in the
        picture?".format,
    "Can you find the {} involved in the accident in the image and give me the coordinates of
        where it is located?".format
]

for object in annotation["accident_objects"]:
    User: random.choice(question_templates)(object["label"]) <image>
    GPT: object['bbox']
```

We define the accident region as the union of all the bounding boxes of the involved objects. The question-answer pair generation for the accident region grounding task is formulated as follows:

```
accident_region = UNION(object["bbox"] for object in annotation["accident_objects"])

question_templates = [
    "Where is the accident?",
    "Where is the accident in the image?",
    "Provide the coordinates of the accident in the image?",
    "Can you point out the accident in the image and provide the coordinates of its location?",
    "Help me to locate the accident in the image and give me its coordinates, please.",
    "In the given image, could you find and tell me the coordinates of the accident?",
    "Guide me to the location of the accident within the image by providing its coordinates.",
    "I'd like to know the exact coordinates of the accident in the photo.",
    "Would you kindly provide the coordinates of the accident located in the picture?",
    "Can you find the accident in the image and give me the coordinates of where it is located?
        "
]

User: random.choice(question_templates) <image>

GPT: accident_region
```

**Single-turn: Accident Recognition and Description.** Here is the pseudo code of generating conversations for the task of accident recognition and description:

```
question_templates = [
   "Is there a traffic accident in the given video clip?",
   "Does this video capture a traffic accident?",
   "Is a traffic accident occurring at any point in this video?",
   "Can you detect any traffic collisions in this video?",
   "Is there evidence of a road traffic accident visible in this video clip?",
   "Throughout this video, is there an incident involving a traffic accident?",
   "Do you observe a car accident happening in the sequence of this video?",
   "Does this video document any vehicular collisions or crashes?",
   "Can you point out if there's a traffic-related accident depicted in this video?",
   "Is there any part of this video that shows a traffic mishap or collision?",
   "Watch this video and confirm if a traffic accident takes place at any moment."
]

User: random.choice(question_templates) <video>

GPT: Yes (for Accident) / No (for Normal). annotation["accident-description"]
```

### B.2 INSTRUCTION TUNING DATA GENERATION

To generate instruction tuning data for traffic accident understanding, we design a set of instructions to guide the general-purpose multimodal language model, such as LLaMA-70B (Dubey et al., 2024) in our work, to generate multi-turn conversations. Trained on the generated multiple rounds of conversations, our TABot is expected to be endowed with a more comprehensive understanding of traffic accidents and equipped with the capability to provide more contextually relevant responses. The statistics of the generated conversation pairs based on our dataset are summarized in Table 11.

Specifically, we follow the in-context-learning (ICL) paradigm of LLaVA (Liu et al., 2024) and adapt it to our traffic accident understanding scenario. The detailed ICL prompting instruction for Llama3 is illustrated in Table 12. In particular, the caption of the video is provided for the model to imagine the visual content of the video, which should be accident-oriented in our work. Therefore, we organize the caption in a structured way: Sentence 1 describes the video source; Sentence 2 describes the accident-related content, in other words, the labeled accident description in the TAU-106K dataset; Sentence 3 complements the accident event with more detailed information, such as the temporal information and the objects involved in the accident.

Besides these complete sentences, we also list the annotated temporal segments and the bounding boxes of the accident-involved objects in the video clips, leading the model to refer to the specific content when generating responses to the user queries. An example of the structured caption and the generated multi-turn conversations are presented in Table 13.

Table 11: Generated conversation pairs based on our TAU-106K.

| Task | Size | Response formatting |
|---|---|---|
| Detection & Description (Image) | 55K | Detect and describe the accident |
| Detection & Description (Video) | 52K | Detect and describe the accident |
| Temporal Localization | 28K | $\{t_{start}, t_{end}\}$ |
| Temporal Referring | 54K | Describe the accident |
| Accident Grounding | 28K | [x0, y0, x1, y1] |
| Object Grounding | 45K | [x0, y0, x1, y1] |
| Complex Conprehension | 70K | Multi-round conversation |
| All | 332K | – |

```
messages = [ {"role":"system", "content": f"'You are an AI visual assistant, and
you are seeing a single video. What you see is provided with a few sentences, describing the same
video you are looking at. Answer all questions as you are seeing the video. The video mainly
focuses on the traffic situation.

In particular, if there is an accident, the timesteps of the accident are provided in the format
of {start_time, end_time} with normalized time values.

In addition, if there is an accident, specific object locations involved in the accident are given,
along with detailed coordinates. The accident region is the area where the accident occurred,
presented as the union of all the bounding boxes of the involved objects. These coordinates are
in the form of bounding boxes, represented as [x1, y1, x2, y2] with floating numbers ranging
from 0 to 1. These values correspond to the top left x, top left y, bottom right x, and bottom
right y.

Design a conversation between you and a person asking about this photo. The answers should be
in a tone that a visual AI assistant is seeing the video and answering the question. Ask diverse
questions and give corresponding answers.

Include questions asking about the visual content of the video, including the object types, counting
the objects, object actions, object locations, relative positions between objects, etc. Only include
questions that have definite answers:
(1) one can see the content in the video that the question asks about and can answer confidently;
(2) one can determine confidently from the video that it is not in the video. Do not ask any question
that cannot be answered confidently.

Also include complex questions that are relevant to the content in the video, for example, asking
about background knowledge of the objects in the video, asking to discuss accidents happening
in the video, reasoning about the possible causes of the accident, analyzing the traffic rules that
have been violated, etc. Again, do not ask about uncertain details. Provide detailed answers when
answering complex questions. For example, give detailed examples or reasoning steps to make the
content more convincing and well-organized. You can include multiple paragraphs if necessary."'}
]
for sample in fewshot_samples:
    messages.append({"role":"user", "content":sample['context']})
    messages.append({"role":"assistant", "content":sample['response']}
    )
messages.append({"role":"user", "content":'\n'.join(query)})
```

Table 12: For each query, we illustrate the prompt construction process for Llama3 to collect `query['response']` from `query['context']`, using few-shot in-context-learning, where examples are from `fewshot_samples`, each example including input `sample['context']` and output `sample['response']`. Note that `messages` is the final prompt. The **bold** text indicates the tailored instruction for the accident-oriented conversation generation. In this illustration, we provide the instruction used to generate the conversation response, please see also see its in-context-learning example in Table 13 for details.

**Captions**
This video is sourced from a traffic surveillance camera.
Surveillance cameras captured the road surface of a rural road, at 0.41 normalized timestamp, a truck collided with an electric rider as it drove straight ahead, at 0.53 normalized timestamp, the owner of the electric bike fell to the ground.
At 0.450 normalized timestamp, a man in a red coat riding an electric bike located at [0.324, 0.301, 0.488, 0.683] and a straight freight car located at [0.472, 0.101, 0.810, 0.849] were involved in the accident.
accident_timesteps: {0.41, 0.53}
accident_objects:
    man in a red coat riding an electric bike: [0.324, 0.301, 0.488, 0.683]
    straight freight car: [0.472, 0.101, 0.810, 0.849]
accident_region: [0.324, 0.101, 0.810, 0.849]

**Conversation**
Question:
Can you describe the types of vehicles involved in the accident captured in the video?
Answer:
The accident involved a freight truck and an electric bike. The freight truck was moving straight ahead on a rural road when it collided with the electric bike.
===
Question:
At what point in the video does the accident start and end?
Answer:
The accident starts at the normalized timestamp of 0.41 and ends at 0.53.
===
Question:
Where exactly did the accident occur in relation to the video frame?
Answer:
The accident occurred within the coordinates [0.324, 0.101, 0.810, 0.849] in the video frame. This region represents the unified area of both involved objects, the electric bike and the freight truck, during the accident.
===
Question:
How did the rider of the electric bike respond to the collision?
Answer:
After the collision, the rider of the electric bike fell to the ground. This incident happened around the 0.53 normalized timestamp in the video.
===
Question:
What is the appearance of the person involved in the accident?
Answer:
The person involved in the accident, specifically the rider of the electric bike, was wearing a red coat. This detail is captured around the 0.450 normalized timestamp in the video when describing their location in the frame.
===
Question:
Can you analyze the possible cause of the accident shown in the video?
Answer:
Based on the video, one possible cause of the accident could be the relative positions and movements of the freight truck and the electric bike. The truck was moving straight ahead on a rural road, suggesting that the visibility or the reaction time could have been compromised for either the truck driver or the electric bike rider. Without specific video evidence of traffic signals or signs, one might infer that perhaps there was a failure to yield the right of way or an issue with speed control. However, definitive causes would require further investigation into the traffic conditions, driver behavior, and environmental factors at the time of the accident.

Table 13: One example used in in-context-learning to construct visual conversation data.

# C   SUPPLEMENTARY ABLATION EXPERIMENTS

**Fine-tuning on Other Baselines.** We extend our experiments to include recent MLLMs, Video-LLaMA-2 (Cheng et al., 2024) and Qwen2-VL (Yang et al., 2024), fine-tuned on our proposed TAU-106K dataset to evaluate its quality and significance. The results, shown in Table 14, demonstrate significant performance improvements across all tasks after fine-tuning, validating the effectiveness of our dataset and training recipes. After fine-tuning, the performance gap between Video-LLaMA-2 and GroundingGPT is reduced, highlighting the importance of fine-tuning on target datasets for accident understanding tasks. Our TABot model based on GroundingGPT, still outperforms fine-tuned Video-LLaMA-2, particularly in image understanding tasks. As for the SOTA model Qwen2-VL, the pre-trained model already achieves competitive performance, and the fine-tuning on TAU-106K further boosts the performance, reaching the highest performance in all tasks. These findings emphasize the necessity of fine-tuning on domain-specific datasets, demonstrating the effectiveness and quality of our comprehensive TAU-106K dataset.

Table 14: The results of fine-tuning other baselines on TAU-106K.

|  | Video Understanding | | | Image Understanding | | |
|---|---|---|---|---|---|---|
|  | CLS (Acc) | TL (AP@50) | CAP (BERT) | CLS (Acc) | AG (AP@50) | CAP (BERT) |
| GroundingGPT | 50.00 | 2.40 | 55.70 | 63.75 | 14.25 | 45.00 |
| + TABot | 81.00 | 20.12 | 82.31 | 90.75 | 70.03 | 75.20 |
| Video-LLaMA-2 | 64.00 | 2.10 | 62.20 | 63.30 | 31.57 | 63.21 |
| + TABot | 79.90 | 19.30 | 83.36 | 77.80 | 57.25 | 73.71 |
| Qwen2-VL | 72.65 | 15.76 | 61.61 | 58.35 | 47.52 | 66.12 |
| + TABot | 82.65 | 22.50 | 83.09 | 92.00 | 77.61 | 76.32 |

**Experiment Results of 7:3 Split.** Since the amount of our TAU-106K dataset is large enough, 1/10 of the data (5K videos and 5K images) is sufficient for testing. However, we have conducted additional experiments with a 7:3 train/test split to verify the model's generalization ability to overfitting, as reported in Table 15. The model's performance remains consistent across different tasks with a slight decrease in accuracy, demonstrating its robustness to different train/test splits.

Table 15: The results of fine-tuning TAU-106K with the 7:3 split.

|  | Video Understanding | | | Image Understanding | | |
|---|---|---|---|---|---|---|
|  | CLS (Acc) | TL (AP@50) | CAP (BERT) | CLS (Acc) | AG (AP@50) | CAP (BERT) |
| TABot (9:1) | 81.00 | 20.12 | 82.31 | 90.75 | 70.03 | 75.20 |
| TABot (7:3) | 79.95 | 19.08 | 81.57 | 88.97 | 68.29 | 74.55 |

**Data Imbalance in Fine-grained Categories.** The class imbalance issue is inevitable in data collection and our data distribution also fairly reflects real-world situations. The primary purpose of this paper is to facilitate the development of MLLM on large-scale traffic datasets and learn models that closely resemble real-world conditions. Besides, we provide Table 16 to show the accuracy of different types of traffic accidents in both video/image accident recognition tasks. While categories like MnMV show slightly lower performance due to the limited amount of training data, the model's performances across different accident types are generally comparable. Overall, the model achieves consistent and satisfactory results across the normal class and the five accident classes.

Table 16: Accident proportion and class-wise accuracy for video and image accident recognition.

| Accident Category | Proportion in TAU-106k | Accuracy for Video Accident Recognition | Accuracy for Image Accident Recognition |
|---|---|---|---|
| Normal | 44.1% | 78.80 | 92.23 |
| SMV | 9.7% | 80.56 | 79.75 |
| MMV | 32.9% | 84.39 | 87.40 |
| MV&nMV | 9.8% | 92.02 | 82.87 |
| V&P | 2.2% | 94.29 | 84.78 |
| MnMV | 1.3% | 77.78 | 78.26 |

**Qualitative Visualization and Comparison.** We provide visualizations and comparisons on the functional image and video understanding tasks in Figure 8, 9, 10, and 11. Besides, the multi-turn dialogue samples on traffic accident videos are presented in Figure 12 and Figure 13.

---

**Question [RD]:** Can you confirm the occurrence of a vehicle accident in this picture? Please respond with a 'Yes' or 'No'. Following that, describe the image in one sentence.

**Ground Truth:** Yes. An electric bicycle rider wearing black collided with a red car making a right turn due to faulty brakes.

**GPT-4o:** Yes, the image shows a traffic accident involving a red car that appears to have collided with another vehicle in a busy intersection.

**Qwen2-VL:** No. The image shows a busy street with multiple cars and a pedestrian crossing the road, but there is no visible accident.

**Gemini-1.5-Pro:** No. A red car is turning right at an intersection in a city.

**TABot-Chat (Ours):** Yes. Because a black-clad electric bicycle rider missed the brakes, it collided with a red car turning right.

---

**Question [SG]:** Where is the traffic accident? The answer should be given in normalized $[x_{min}, y_{min}, x_{max}, y_{max}]$ format.

**Ground Truth:** [0.310, 0.343, 0.881, 0.935]
**Qwen2-VL:** [0.31, 0.35, 0.94, 0.97]
**Gemini-1.5-Pro:** [0.38, 0.24, 0.98, 0.94]
**TABot-Chat (Ours):** [0.321, 0.319, 0.878, 0.937]

Figure 8: One qualitative example of the functional image understanding tasks.

---

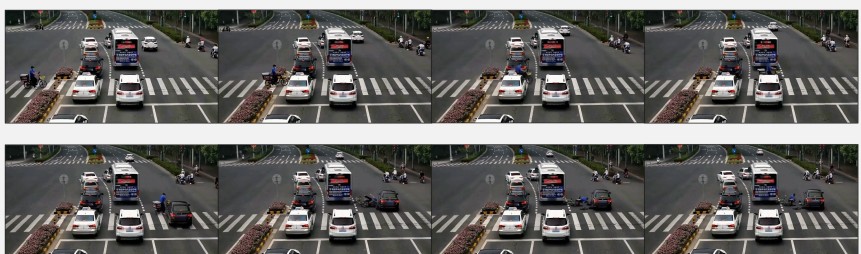

**Question [RD]:** Does this video document any vehicular collisions or crashes? Please respond with a 'Yes' or 'No'. Following that, describe the video in one sentence.

**Ground Truth:** Yes. The surveillance camera showed a black car colliding with a blue-clad electric bike rider at a city intersection, knocking the rider to the ground.

**Qwen2-VL:** Yes. A motorcycle is hit by a car and the rider falls to the ground.

**Gemini-1.5-Pro:** Yes. A cyclist is hit by a car at a crosswalk.

**TABot-Chat (Ours):** Yes. Surveillance cameras captured the road surface of a city street, as a black car ran through the intersection, it collided with an electric bicycle rider dressed in blue, the cyclist was knocked to the ground.

---

**Question [TL]:** Show me when the traffic accident gets going and when it's all over. The answered timestamp should be formatted in a normalized manner, using {start_time, end_time}.

**Ground Truth:** {0.55, 0.86}
**Qwen2-VL:** {0.50, 0.80}
**Gemini-1.5-Pro:** {0.80, 0.88}
**TABot-Chat (Ours):** {0.56, 0.84}

Figure 9: One qualitative example of the functional video understanding tasks.

**Question [RD]:** Are there any indications of a traffic accident in this image? Please respond with a 'Yes' or 'No'. Following that, describe the image in one sentence.

**Ground Truth:** Yes. A blue car did not pay attention to the road conditions while making a left turn, which caused the current vehicle to take evasive action, leading to a collision between the blue car and the current vehicle. Subsequently, both cars crashed into the left road guardrail.

**GPT-4o:** Yes, the image shows a car colliding with a barrier while another vehicle is in the foreground.

**Qwen2-VL:** No. The image shows a car driving on a road with other vehicles turning left and traffic signs visible.

**Gemini-1.5-Pro:** No. The image shows a car driving on a highway, approaching a bridge.

**TABot-Chat (Ours):** Yes. Because a blue car did not notice the condition of the road when turning left, causing the current vehicle to avoid it, the current vehicle collided with the blue car, then the blue car and the current vehicle collided with a fence on the left side of the road.

---

**Question [SG]:** In the given image, could you tell me the coordinates of the blue car recorded in crash incidents? The answer should be given in normalized $[x_{min}, y_{min}, x_{max}, y_{max}]$ format.

**Ground Truth:** [0.392, 0.275, 0.998, 0.956]
**Qwen2-VL:** [0.38, 0.38, 1.0, 0.66]
**Gemini-1.5-Pro:** [0.36, 0.28, 0.88, 0.82]
**TABot-Chat (Ours):** [0.396, 0.283, 1.000, 0.939]

Figure 10: One qualitative example of the functional image understanding tasks.

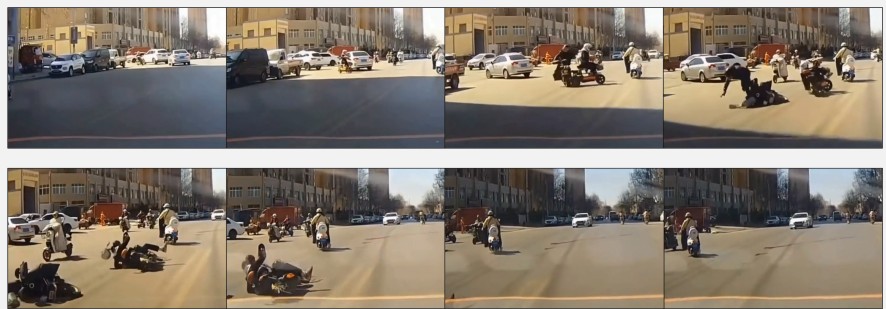

**Question [RD]:** Do you observe a car accident happening in the sequence of this video? Please respond with a 'Yes' or 'No'. Following that, describe the video in one sentence.

**Ground Truth:** Yes. A vehicle was driving on a city street when two electric bicycle riders collided because one, dressed in dark clothing, failed to brake in time. The other rider, wearing a white helmet, was also struck, and both fell to the ground.

**Qwen2-VL:** Yes. A motorcycle falls down on the road in the video.

**Gemini-1.5-Pro:** Yes. A dashcam video shows a motorcyclist falling from their vehicle after colliding with another vehicle on a city street.

**TABot-Chat (Ours):** Yes. The vehicle was traveling on the streets of the city, because an electric bicycle rider in dark clothes did not brake in time, it collided with a white helmeted electric bicycle rider, both riders fell to the ground.

---

**Question [TL]:** Can you delineate the duration of the traffic accident from beginning to end? The answered timestamp should be formatted in a normalized manner, using {start_time, end_time}.

**Ground Truth:** {0.39, 0.75}
**Qwen2-VL:** {0.25, 0.75}
**Gemini-1.5-Pro:** {0.80, 0.95}
**TABot-Chat (Ours):** {0.39, 0.72}

Figure 11: One qualitative example of the functional video understanding tasks.

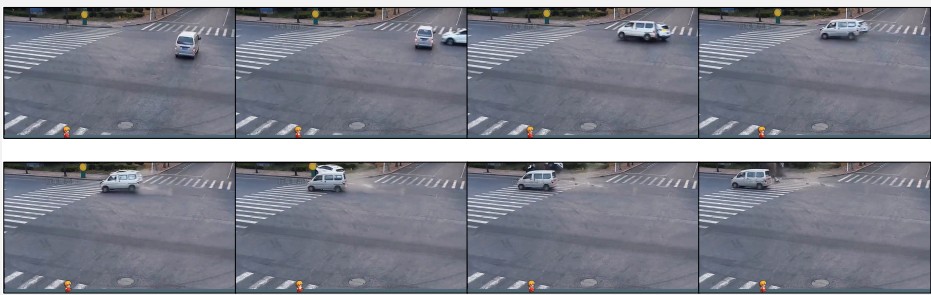

**User:** How many vehicles were involved in the accident captured by the traffic surveillance camera?
**TABot-Chat:** There were two vehicles involved in the accident: a silver bread truck and a white car. Additionally, the white car also crashed into a sign on the side of the road.

**User:** What is the duration of the accident as seen in the video?
**TABot-Chat:** The accident starts at the normalized timestamp of 0.12 and ends at 1.0, indicating a relatively long duration.

**User:** Can you describe the sequence of events in the accident?
**TABot-Chat:** The accident unfolded in two stages. Initially, at 0.218 normalized timestamp, the white car collided with the silver bread truck at the intersection. Later, at 0.609 normalized timestamp, the same white car crashed into a sign on the side of the road.

**User:** Where did the accident occur in relation to the video frame?
**TABot-Chat:** The accident occurred within the region bounded by the coordinates [0.317, 0.000, 0.941, 0.279] in the video frame. This region encompasses both collision scenes, including the intersection where the white car hit the silver bread truck and the area where the white car crashed into the sign.

**User:** What can be inferred about the white car's behavior during the accident?
**TABot-Chat:** Based on the video, it appears that the white car was involved in two separate collisions. The first collision occurred when it hit the silver bread truck, and the second collision happened when it crashed into the sign on the side of the road. This sequence of events suggests that the white car may have lost control or failed to stop after the initial collision, leading to the subsequent crash into the sign.

**User:** Can you provide any insights into the possible causes of the accident?
**TABot-Chat:** The video evidence suggests that the white car may have been traveling at an excessive speed or failed to obey traffic rules, leading to the initial collision with the silver bread truck. The subsequent crash into the sign could be attributed to the driver's loss of control or failure to react appropriately after the initial collision. However, without additional information, such as the speed of the vehicles, road conditions, or driver behavior, a definitive cause cannot be determined solely from the footage.

Figure 12: One qualitative example of the multi-turn dialogue on the traffic accident video.

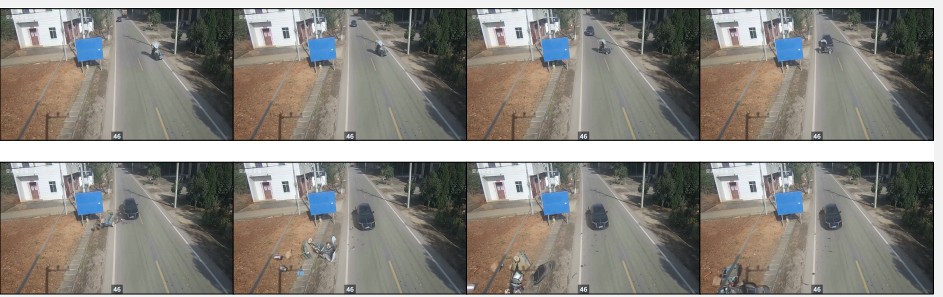

**User:** What type of vehicles were involved in the accident captured by the traffic surveillance camera?
**TABot-Chat:** The accident involved a black car and a blue tricycle. The black car was moving on the country road when it collided with the blue tricycle, which was turning left.

**User:** What was the cause of the accident according to the video?
**TABot-Chat:** The accident occurred because the black car failed to brake on time, resulting in a collision with the blue tricycle.

**User:** At what point in the video does the accident start and end?
**TABot-Chat:** The accident starts at the normalized timestamp of 0.30 and ends at 0.89.

**User:** Where exactly did the accident occur in relation to the video frame?
**TABot-Chat:** The accident occurred within the coordinates [0.490, 0.223, 0.582, 0.410] in the video frame. This region represents the unified area of both involved objects, the blue tricycle and the black car, during the accident.

**User:** What happened to the tricycle and the cyclist as a result of the accident?
**TABot-Chat:** The tricycle overturned, and both the tricycle and the cyclist were hit from the road surface.

**User:** Can you analyze the possible consequences of the accident shown in the video?
**TABot-Chat:** Based on the video, the consequences of the accident could be severe. The tricycle overturning and the cyclist being hit from the road surface suggest potential injuries to the cyclist. The accident could also have caused damage to both vehicles involved. Additionally, the accident may have caused a disruption to traffic flow on the country road. A thorough investigation would be necessary to determine the full extent of the consequences.

**User:** What traffic rule might have been violated by the black car driver?
**TABot-Chat:** The black car driver might have violated the rule of maintaining a safe following distance or failing to yield to the blue tricycle, which was turning left. The driver's failure to brake on time suggests a possible failure to exercise due care and caution while driving.

Figure 13: One qualitative example of the multi-turn dialogue on the traffic accident video.

