# OpenReview forum: "TAU-106K: A New Dataset for Comprehensive Understanding of Traffic Accident"
_ICLR.cc/2025/Conference — ICLR 2025 Poster_

### Official Review · Reviewer_Vxed · 2024-10-28

**Soundness:** 3
**Presentation:** 2
**Contribution:** 2
**Rating:** 6
**Confidence:** 3

**Summary:**

This paper proposes a new dataset, TAU-106K, to help MLLMs gain a comprehensive understanding of traffic accidents. In addition, the paper also proposes a two-stage fine-tuning mechanism to use the proposed dataset. The experimental results basically prove the effectiveness of the proposed method and data.

**Strengths:**

The dataset proposed in this paper covers a wide range of knowledge about traffic accidents, which has greatly promoted the development of MLLMs in this specific field. The diversity and details of the proposed dataset are meaningful and meet the expectations of MLLMs to learn specific knowledge.

**Weaknesses:**

This paper provides a meaningful dataset and benchmark, but I still have some concerns:

* Although the experimental results basically prove the superiority (table 1, 2, 3, 4), the highlighted information is confusing: Is the performance improvement due to the rich knowledge of the data or the fine-tuning strategy? Both aspects should have an effect, so there should be more comparisons and ablations. For example, you can use your proposed dataset to fine-tune other baseline models.

* Accident Recognition uses the Yes and No responses of the model to make binary classifications. In a random situation, are the probabilities of the model responding Yes and No balanced? If not, will this cause the evaluation to be disturbed by some irrelevant prior knowledge? You can start from reporting the base rates of Yes/No responses.

* There are some spelling and grammatical errors that the author would like to note (for example: line 192 'hey', line 238 'is', line 375 'a the', ...).

**Questions:**

The author can think more about the proposed data and benchmarks:

* In addition to the defined traffic accidents, there are actually more conner cases, which are often the most difficult to learn in the open world. How to use the proposed method to let MLLMs learn this knowledge?

* Is there greater potential for the generalization of data?

---

> ### Author Response · Authors · 2024-11-24
> **Response to Reviewer Vxed**
>
> >**[Effectiveness of Dataset vs. Fine-tuning Strategy]:** Is the performance improvement due to the rich knowledge of the data or the fine-tuning strategy? Both aspects should have an effect, so there should be more comparisons and ablations, such as fine-tuning other baseline models with the proposed dataset.
>
> **Answer:**
> In our work, the TAU-106K dataset is the primary contribution, while the proposed components are designed to demonstrate the dataset’s utility and facilitate future research.
> On the other hand, our proposed training recipes (Video Spatial Alignment and Negative Segment Referring) have shown consistent improvements over the baseline model, indicating their effectiveness in enhancing accident-aware video understanding.
>
> Additionally, we have fine-tuned other baseline models on the TAU-106K dataset to evaluate the impact of the dataset on generalization performance.
> In detail, the very recent MLLMs, Video-LLaMA-2 and Qwen2-VL, are selected as the alternative baselines for comparison.
> Here we report the results of accident understanding tasks after fine-tuning the TAU-106K dataset in the following table:
>
> | Model         | V-CLS (Acc) | V-TL (AP@50) | V-CAP (BERT) | I-CLS (Acc) | I-AG (AP@50) | I-CAP (BERT) |
> | ------------- | ----------- | ------------ | ------------ | ----------- | ------------ | ------------ |
> | GroundingGPT  | 50.0        | 2.4          | 55.7         | 63.8        | 49.1         | 45.0         |
> | + TABot       | 81.0        | 20.1         | 82.3         | 90.8        | 65.9         | 75.2         |
> | Video-Llama-2 | 64.0        | 2.1          | 62.2         | 63.3        | 31.6         | 63.2         |
> | + TABot       | 79.9        | 34.3         | 83.4         | 77.8        | 57.3         | 73.7         |
> | Qwen2-VL      | 72.7        | 15.8         | 61.6         | 58.4        | 47.5         | 66.1         |
> | + TABot       | 82.7        | 22.5         | 83.1         | 92.0        | 77.6         | 76.3         |
>
> When fine-tuned on our TAU-106K dataset, the Video-LLaMA-2 and Qwen2-VL models achieve a significant improvement in all tasks, demonstrating the effectiveness of our proposed training recipes and the quality of our dataset.
> Notably, the CAP performance of Video-LLaMA-2 is even better than our proposed TABot model, and we conjecture that the performance gap is due to the enhanced captioning capabilities of Video-LLaMA-2 pre-trained on a larger-scale general-purpose dataset.
> After fine-tuning, the performance gap between Video-LLaMA-2 and GroundingGPT is narrowed, which indicates the importance of fine-tuning on the target dataset and the specificity of accident understanding tasks.
> In particular, the overall performance of our proposed TABot model based on GroundingGPT is still superior to fine-tuned Video-LLaMA-2, especially in image understanding tasks.
> As for the SOTA model Qwen2-VL, the pre-trained model already achieves competitive performance, and the fine-tuning on TAU-106K further boosts the performance, reaching the highest performance in all tasks.
> We conjecture that the competitive performance of the pre-trained Qwen2-VL is due to its advanced architecture and pre-training strategies, and maybe some accident-related or traffic-related data in the pre-training dataset.
> Even so, the performance improvement after fine-tuning on TAU-106K demonstrates the necessity of fine-tuning on the domain-specific dataset, highlighting the effectiveness and quality of our comprehensive TAU-106K dataset.
>
> ---
>
>
> >**[Bias in Yes/No Responses for Accident Recognition]:** Accident Recognition uses Yes/No responses for binary classification. In random situations, are the probabilities of Yes/No responses balanced? If not, will this bias evaluations due to irrelevant prior knowledge?
>
> **Answer:**
> In our TAU-106K dataset, 56% of the instances are labeled as "Accident," and 44% as "Normal," resulting in a relatively balanced Yes/No distribution overall. Specifically, in the test set for the video data, which consists of 2,000 video samples, 50% are positive samples (Yes) and 50% are negative samples (No). The model produces 53.2% Yes responses and 46.8% No responses. This indicates that the model's responses to the video data are balanced, showing no significant bias when performing video understanding tasks.
>
> The test set of the image data also contains 2,000 samples, with 55.2% positive samples (Yes) and 44.8% negative samples (No). The model outputted 61.4% Yes responses and 38.6% No responses. Overall, the model shows a very slight bias toward positive samples, but this bias is minimal and remains within an acceptable range. We understand that this is likely due to a slight imbalance in the training data, which is reflected in the test results.
>
> ---
>
> **Additional responses are left to the next comment window.**

---

> > ### Author Response · Authors · 2024-11-24
> > **Response to Reviewer Vxed**
> >
> > >**[Clarity in Writing]:** There are some spelling and grammatical errors, such as "hey" (line 192), "is" (line 238), and "a the" (line 375).
> >
> > **Answer:** Thank you for pointing it out. We will carefully proofread the manuscript and correct all identified spelling and grammatical errors to ensure clarity and professionalism in the writing.
> >
> > ---
> >
> > >**[Corner Cases and Open-world Learning]:** How can the proposed method enable MLLMs to learn about corner cases, which are often the hardest to address in the open world?
> >
> > **Answer:**
> >
> > | Accident Category | Proportion | Acc@Video | Acc@Image |
> > | ----------------- | ---------- | --------- | --------- |
> > | Normal            | 44.1%      | 78.80     | 92.23     |
> > | SMV               | 9.7%       | 80.56     | 79.75     |
> > | MMV               | 32.9%      | 84.39     | 87.40     |
> > | MV&nMV            | 9.8%       | 92.02     | 82.87     |
> > | V&P               | 2.2%       | 94.29     | 84.78     |
> > | MnMV              | 1.3%       | 77.78     | 78.26     |
> >
> > Thank you for your thoughtful feedback. Corner cases are indeed challenging for models to learn effectively. In our dataset, however, we did not apply any specific adjustments to handle these cases. Instead, we designed the dataset to reflect the real-world distribution of different types of accidents, aiming to create a more realistic simulation.
> >
> > From the results, we can see that although categories like MnMV and V&P have a relatively small proportion in the dataset, the model still performs reasonably well in these categories. Notably, V&P achieves very high accuracy, while MnMV has slightly lower accuracy but still delivers satisfactory performance. This demonstrates that our model can make effective predictions even for corner cases. To address the issues associated with extremely limited corner case data, potential future solutions could include generative data augmentation, data resampling, or incorporating new external data.
> >
> > ---
> >
> > >**[Generalization Potential of Data]:** Is there greater potential for the generalization of the dataset beyond the defined traffic accident scenarios?
> >
> > **Answer:**
> > While TAU-106K is specifically designed for traffic accident understanding, it also possesses the potential for generalization to other tasks.
> >
> > 1. Various data from urban, highway, and rural environments make it suitable for understanding complex dynamic scenes, such as robotics navigation, autonomous driving, and surveillance-based analysis.
> >
> > 2. TAU-106K’s strengths in temporal localization, spatial grounding, and description generation provide a foundation for fine-grained scene understanding like disaster response, human activity recognition, and other applications.
> > However, as the dataset’s conversational paradigm is tailored to traffic accident scenarios, users may need to fine-tune the data structure or adapt it to better suit their specific application domains.
> >
> > 1. The multimodal annotations of TAU-106K (video, image, and textual descriptions) can be used as part of the training set for multi-modal reasoning tasks. These tasks include causal reasoning (inferring the causes and consequences of events across various domains), scene understanding (generating detailed textual explanations of dynamic environments), and event localization and explanation (accurately identifying when and where significant events occur and explaining them in natural language).
> >
> > In summary, TAU-106K may offer significant potential to support interdisciplinary research and diverse applications with minor adjustments to its structure for broader use cases and applications.

---

> ### Author Response · Authors · 2024-11-30
> **Dear Reviewer Vxed**
>
> Dear **Reviewer Vxed**,
>
> We would greatly appreciate it if you could review our response by **Dec 1 AoE**. After that date, it might be challenging for us to engage in further discussions.
>
> If you have any follow-up questions, please don't hesitate to reach out. We deeply value your expertise and time.
>
> Best regards,
> The authors of *TAU-106K: A New Dataset for Comprehensive Understanding of Traffic Accident*

---

> > ### Comment · Reviewer_Vxed · 2024-11-30
> > **Thanks for your response**
> >
> > Thank you for your reply. I think this paper is a sufficient contribution to the conference and the related domain, I will keep my score.

---

> > > ### Author Response · Authors · 2024-12-02
> > > **Thanks for your recognition**
> > >
> > > Dear Reviewer Vxed,
> > >
> > > Thank you for your kind and encouraging feedback. We deeply appreciate your recognition of our work and your valuable time in reviewing our paper. Your positive comments motivate us to further refine our research and continue contributing to the field.
> > >
> > > Best regards,
> > > The authors of *TAU-106K: A New Dataset for Comprehensive Understanding of Traffic Accident*

---

### Official Review · Reviewer_1r9r · 2024-10-31

**Soundness:** 3
**Presentation:** 2
**Contribution:** 3
**Rating:** 6
**Confidence:** 5

**Summary:**

This paper proposes a large-scale multimodal dataset, TAU-106K, aimed at addressing the limitations of multimodal large language models (MLLMs) in high-level and fine-grained anomaly understanding, specifically for traffic accident scenarios. The TAU-106K dataset comprises images and videos sampled from existing datasets and retrieved from online sources, along with annotated text data. The manuscript also proposed a two-stage fine-tuning approach named TABot to integrate multi-level accident understanding tasks, including accident recognition, spatiotemporal grounding, and an additional accident description task to further enhance the model’s understanding of traffic accidents. While TAU-106K shows potential as a valuable resource, the paper would benefit from a clearer elaboration on the unique challenges of the traffic accident domain and additional experimental comparisons.

**Strengths:**

This paper focuses on the important field of traffic accident scene understanding, addressing the current limitations of large models in high-level and fine-grained semantic comprehension. It constructs large-scale multimodal models, with a particular emphasis on anomalies, which are critical focal points in the field of autonomous driving. The introduction of this dataset is expected to further promote advancements in this area.
The proposed dataset is designed for various tasks related to traffic accident scenes, including accident recognition, accident description, temporal localization, and spatial grounding. It also features multi-granularity annotations for the collected images and videos, effectively aiding the model in further learning and understanding of traffic accidents.

**Weaknesses:**

The presentation of the dataset requires further optimization, as the focus on traffic accidents carries significant practical implications and research value. However, the paper lacks clarity regarding the challenges associated with understanding traffic accidents, and the characteristics of such incidents need to be more precisely articulated.
While the two-step fine-tuning approach is widely recognized, the distinctions between the proposed method and established general methods are not clearly delineated. This creates a disconnect between the methodology and the traffic accident focus of the dataset. Furthermore, the method’s design does not sufficiently address specific tasks, necessitating clearer articulation of its innovations and contributions.
In the experimental section, there is a need for more comprehensive validation of the challenge of dataset and the effectiveness of method, particularly concerning experimental setup and evaluation metrics.

**Questions:**

1. How does this dataset compare with existing related datasets? Can specific data be provided? Are there any additional benefits beyond language descriptions? What sets it apart from traditional VQA tasks?
2. Could the annotation process from video to image be more detailed?  And how is multi-turn dialogue accuracy ensured? What steps are taken to ensure annotation reliability, efficiency, and cost-effectiveness?
3. Anomalous events typically show a long-tail distribution. Is this characteristic present in the dataset? How are the resulting distributions in the training and test sets structured?
4. Instruction tuning is classic and common. What is the two-step fine-tuning approach that differs from previous methods? How does Video Spatial Alignment differ specifically from GroundingGPT? Besides input data differences, what specific innovations are presented? Negative Segment Referring is vaguely explained—does it refer to non-accident samples, and how does it differ from standard practices? Hybrid data and task markers are often seen in model fine-tuning; what is their connection to traffic accident detection?
5. The experimental section and method descriptions aren’t well aligned. For instance, what are the distinctions between Negative Segment Referring, multi-turn dialogues, TABot (Video), TABot (Ours), and TABot-Chat (Ours)? Does Ours use both video and image training data? How is multi-turn dialogue in -Chat specifically designed? More clarity is needed.
6. What is the relationship between the four tasks: accident recognition, accident description, temporal localization, and spatial grounding? Could these tasks validate each other to improve model performance? What is Image Understanding in the experiment section, and how does it relate to the other four tasks?
7. The authors achieved remarkable results with GroundingGPT. How is the performance of the proposed method based on other models?
8. For some metrics, GroundingGPT achieves 100% accuracy, while for others, it scores 0%. What causes this drastic disparity, and could it be related to experiment settings or metric choices?
9. In the case study, how do prediction results from different methods compare to ground-truth annotations? Additionally, some details need review. For instance, the conclusion mentions an image count of 55k, which seems inconsistent with earlier descriptions.

---

> ### Author Response · Authors · 2024-11-24
> **Response to Reviewer 1r9r**
>
> >**[Clarity on Dataset Challenges and Characteristics]:** The paper lacks clarity regarding the challenges associated with understanding traffic accidents and the characteristics of such incidents. How does this dataset compare with existing related datasets, and what specific benefits does it offer beyond language descriptions?
>
> **Answer:**
> As illustrated in Figure 1 of the manuscript, accident, a typical anomalous event in traffic scenes, is a complex and high-level concept that is challenging to the general-purpose MLLMs to understand.
> The challenges of traffic accident understanding include complex temporal dynamics, specific visual patterns, and task-oriented descriptions that require a deep understanding of the traffic context.
> Therefore, we construct TAU-106K, a large-scale dataset with fine-grained annotations including pre-existed labels (accident categories, temporal annotations, and object grounding) and the additional manually annotated free-form captions, to facilitate research on traffic accident understanding and related tasks, to facilitate research on traffic accident understanding.
>
> Here we provide a comparison of TAU-106K with other accident-specific or general-purpose benchmarks in terms of size, annotation types, supported tasks, and caption diversity:
>
> | Dataset              | Years | Domain           | # Videos | Annotations                                                                 | # Words per caption | Avg. Duration |
> | -------------------- | ----- | ---------------- | --------- | --------------------------------------------------------------------------- | -------------------- | ------------- |
> | Dashcam              | 2016  | Traffic          | 3,000     | Temporal;                                                                   | -                    | 5.0 seconds   |
> | A3D                  | 2019  | Traffic          | 1,500     | Temporal;                                                                   | -                    | 8.5 seconds   |
> | CCD                  | 2021  | Traffic          | 1,500     | Temporal;                                                                   | -                    | 5.0 seconds   |
> | TAD                  | 2021  | Traffic          | 500       | Accident Categories; Temporal;                                              | -                    | 35.8 seconds  |
> | DADA                 | 2021  | Traffic          | 200       | Accident Categories; Temporal; Driver Attention                             | -                    | 11.0 seconds  |
> | SUTD-TrafficQA       | 2021  | Traffic          | 10,080    | QA pairs                                                                    | -                    | 13.6 seconds  |
> | DoTA                 | 2022  | Traffic          | 4,677     | Accident Categories; Temporal; Accident-involved Objects                    | -                    | 15.6 seconds  |
> | CAP                  | 2023  | Traffic          | 11,727    | Accident Categories; Temporal; Fixed-Form Caption                           | 6.3 words            | 6.2 seconds   |
> | TAD-106K             | 2024  | Traffic          | 51,544    | Accident Categories; Temporal; Accident-involved Objects; Free-Form Caption | 32.1 words           | 10.3 seconds  |
> | Charades-STA         | 2017  | Daily activities | 9,848     | Temporal; Free-Form Caption                                                 | 6.3 words            | 31 seconds   |
> | DiDeMo               | 2017  | Open             | 10,464    | Temporal; Free-Form Caption                                                 | 7.5 words            | 30 seconds   |
> | ActivityNet-Captions | 2017  | Open             | 19,209    | Temporal; Free-Form Caption                                                 | 13.5 words           | 180 seconds  |
>
> According to the comparison, our TAU-106K is the largest dataset in terms of the number of videos and the variety of annotations, supporting a wide range of tasks including accident detection, temporal localization, accident grounding, and accident comprehension.
> In particular, benefiting from our manual annotation process that is labor-intensive yet worthy, the labeled free-form accident captions in TAU-106K are much more diverse and detailed than other datasets, achieving a largest average length of 32.1 words per caption. This makes TAU-106K a valuable resource for training and evaluating accident-aware models in traffic video understanding.
>
> ---
> **Additional responses are left to the next comment window.**

---

> > ### Author Response · Authors · 2024-11-24
> > **Response to Reviewer 1r9r**
> >
> > >**[Details on Annotation Process]:** Could the annotation process from video to image be more detailed? What steps are taken to ensure annotation reliability, efficiency, and cost-effectiveness?
> >
> > **Answer:**
> > We use an internal annotation tool to enable interactive use with our annotaters and a diagram of the annotation protocol used in our data engine, which is illustrated in the **Figure 8 (Page 14) of the upgraded supplementary material PDF file**. Additionally, we provide detailed illustrations of the annotations for each accident types in the dataset, which are shown in the **Figure 9 (Page 15) of the upgraded supplementary material PDF file**.
> > As we introduced in the manuscript, the annotation process is divided into two parts: video-based annotation and image-based annotation. Each part is separated into annotation and verification phases carried out by different annotators. The annotators in the verification phases are tasked with verifying the quality of each data item as "satisfactory" or "unsatisfactory". Unsatisfactory items were sent back to the annotation pipeline for refinement.
> >
> > 1. Video Annotation: Annotator A provides accident categories, timestamps, and detailed descriptions for accident videos. Annotator B focuses on quality verification.
> > 2. Image Annotation: In the image annotation phase, accident categories and detailed descriptions are extracted from video annotation. As for the accident-involved objects grounding labels, annotator C first selects keyframes from the candidate keyframes that are averagely sampled according to the labeled accident timestamps. In detail, 9 candidate frames are sampled, and annotator C selects the keyframe that is most representative of the annotated accident description. Then, annotator C grounds the accident-involved objects in the keyframe, labeling the grounded objects as the phases extracted from the accident description. For example, if the description is "A blue car collides with a pedestrian in white clothes", the annotator will only ground the accident-involved car and pedestrian and name them as "blue car" and "pedestrian in white clothes". Annotator D conducts the image-level quality verification.
> >
> > In particular, when the annotator C is grounding the accident-involved objects, the annotator C is asked to select the keyframe according to the accident description and temporal information, which is also a verification process for the video annotation. The annotator D will verify the quality of the image annotation, which is also a verification process for the accident categories and detailed description in the video annotation. Overall, the video and image annotation phases are closely related and mutually verified, ensuring the high quality of the annotation data.
> >
> > ---
> >
> > >**[Long-tail Distribution of Anomalous Events]:** Anomalous events often follow a long-tail distribution. Is this characteristic present in the dataset? How are the training and test sets structured regarding this distribution?
> >
> > **Answer:**
> > In our data collection process, we first gather accident-related data as much as possible, and then collect a similar number of normal samples to keep a balanced distribution: 56% for anomalous samples and 44% for normal. This ensures that the dataset provides sufficient accident-related examples for training and evaluation.
> > As for the training and test set split, we keep the same distribution of normality in the training and test sets to generate a fair evaluation environment for the models.
> >
> > Here we additionally report the distribution of accident categories in the TAU-106K dataset:
> >
> > | Accident Category | Proportion | Acc@Video | Acc@Image |
> > | ----------------- | ---------- | --------- | --------- |
> > | Normal            | 44.1%      | 78.80     | 92.23     |
> > | SMV               | 9.7%       | 80.56     | 79.75     |
> > | MMV               | 32.9%      | 84.39     | 87.40     |
> > | MV&nMV            | 9.8%       | 92.02     | 82.87     |
> > | V&P               | 2.2%       | 94.29     | 84.78     |
> > | MnMV              | 1.3%       | 77.78     | 78.26     |
> >
> > The detailed accident category distribution imbalance issue is inevitable in data collection and our data distribution also fairly reflects real-world situations. The primary purpose of this paper is to facilitate the development of MLLM on large-scale traffic dataset and learn models that closely resemble real-world conditions. Besides, we provide a table showing the accuracy of different types of traffic accidents in both video/image accident recognition tasks. While categories like MnMV show slightly lower performance due to the limited amount of training data, the model's performances across different accident types are generally comparable. Overall, the model achieves consistent and satisfactory results across the normal class and the five accident classes.
> >
> > ---
> > **Additional responses are left to the next comment window.**

---

> > > ### Author Response · Authors · 2024-11-24
> > > **Response to Reviewer 1r9r**
> > >
> > > >**[Two-step Fine-tuning and Methodological Innovations]:** Instruction tuning is classic and common. What is the two-step fine-tuning approach that differs from previous methods? How does Video Spatial Alignment differ specifically from GroundingGPT? Besides input data differences, what specific innovations are presented? Negative Segment Referring is vaguely explained—does it refer to non-accident samples, and how does it differ from standard practices? Hybrid data and task markers are often seen in model fine-tuning; what is their connection to traffic accident detection?
> > >
> > > **Answer:**
> > >
> > > *(i) Instruction tuning is classic and common. What is the two-step fine-tuning approach that differs from previous methods?*
> > >
> > > We indeed adopted a standard fine-tuning framework but made specific adjustments for our scenario.
> > >
> > > 1. During the functional tuning stage, we utilized pre-structured data annotations to populate our QA pairs. This allowed us to prioritize optimizing the functional metrics for the four downstream tasks related to traffic accident recognition and understanding.
> > >
> > > 2. During the instruction tuning stage, we employed the common strategy of expanding the data based on open-source models. However, it is worth noting that we deliberately avoided using state-of-the-art multimodal models for instruction tuning. The reason is that multimodal models, when expanding the data, often generate descriptions based on their own interpretation of visual data. Our TAU-106K dataset, on the other hand, is entirely human-annotated, ensuring a high level of accuracy in the data. Introducing MLLM-generated data could corrupt the annotation quality, especially for fine-grained tasks such as temporal localization and spatial grounding. Therefore, we relied solely on the language-only LLaMA-70B for expanding the QA pairs in our dataset, focusing on improving the conversational capabilities of our model, TABot, without compromising the quality of our annotations.
> > >
> > > *(ii) How does Video Spatial Alignment differ specifically from GroundingGPT?*
> > >
> > > Video Spatial Alignment is our novelty in dialogue data construction. GroundingGPT’s training data for image tasks emphasizes image understanding and spatial grounding, while for video tasks, it only focuses on video understanding and temporal localization. However, we argue that spatial understanding is equally important when performing video understanding and localization. As discussed in Section 4.1 of the paper, we created QA data that links video understanding with spatial understanding. This approach led to significant performance improvements when compared against baseline methods.
> > >
> > >
> > > *(iii) Besides input data differences, what specific innovations are presented?*
> > >
> > > Our work mainly focuses on the proposed dataset and related training recipes in constructing the training dialogue data. In our work, the TAU-106K dataset is the primary contribution, while the proposed components are designed to demonstrate the dataset's utility and facilitate future research.
> > > On the other hand, our proposed training recipes (Video Spatial Alignment and Negative Segment Referring) have shown consistent improvements over the baseline model, indicating their effectiveness in enhancing accident-aware video understanding.
> > >
> > > *(iv) Negative Segment Referring is vaguely explained—does it refer to non-accident samples, and how does it differ from standard practices?*
> > >
> > >
> > > Conventional temporal localization tasks typically involve two types of QA formats:
> > >
> > > * The question asks the model to provide the temporal localization boundaries of the accident.
> > >
> > > * The question provides a specific time interval and asks the model to judge whether a traffic accident occurred within that interval.
> > >
> > > In type 2, typical temporal localization tasks usually contain only positive samples (segments containing accidents) for training. In contrast, when we construct our type 2 data, we include negative samples in the questions, which are normal time intervals without accidents, and include this part of the data for training. By explicitly asking the model to differentiate between these normal segments and accident segments, we enhanced the model's ability to identify traffic accidents more effectively.
> > >
> > > ---
> > > **Additional responses of the sub-question ''Hybrid data and task markers are often seen in model fine-tuning; what is their connection to traffic accident detection?'' are left to the next comment window.**

---

> > > > ### Author Response · Authors · 2024-11-24
> > > > **Response to Reviewer 1r9r**
> > > >
> > > > >**[Two-step Fine-tuning and Methodological Innovations]:** Hybrid data and task markers are often seen in model fine-tuning; what is their connection to traffic accident detection?
> > > >
> > > > **Hybrid Data**: In our task, the purpose of hybrid data design is to address the potential catastrophic forgetting in our two-step fine-tuning framework. The second stage, instruction tuning, can potentially degrade the functional performance of the four downstream tasks (e.g., image accident recognition, image spatial grounding, video accident recognition, and video temporal localization) compared to the first stage. This is similar to the catastrophic forgetting phenomenon. To mitigate this issue, we mixed the data from both stages (similar to adding rehearsal data in continual learning) and used it for instruction tuning. This approach ensures that the model shall improve its conversational capabilities and also retain its functional performance across the four downstream tasks.
> > > >
> > > > **Task Flag**: The employed task flags are a widely used technique for multi-task LLMs/MLLMs. In our task accident scenarios, our model is capable of addressing three different kinds of tasks in traffic video understanding (i.e., RD: accident recognition & description, SG: image spatial grounding, TL: video temporal localization), and the response of the model shall be completely different when coming to these three kinds of tasks. To enable the model to respond appropriately to different tasks, we incorporated a task flag into the questions. This allowed the model to produce task-specific responses for each task.
> > > >
> > > >
> > > > ---
> > > >
> > > > >**[Task Relationships and Validation]:** What is the relationship between the four tasks (accident recognition, accident description, temporal localization, spatial grounding)? Could they validate each other to improve model performance?
> > > >
> > > > **Answer:**
> > > > Accident recognition and accident description are two fundamental tasks, shared by both the video understanding and image understanding tasks.
> > > > As for the specific tasks in each modality: temporal localization in video understanding, and spatial grounding in image understanding, they are complementary to each other, which is validated by the consistent improvement of the model performance when combining these tasks.
> > > > In detail, as shown in Table 1-4 of the manuscript, the performance of TABot (Ours) outperforms TABot (Video) or TABot (Image) on all tasks, which are trained with only video or image data respectively.
> > > >
> > > > ---
> > > > **Additional responses are left to the next comment window.**

---

> > > > > ### Author Response · Authors · 2024-11-24
> > > > > **Response to Reviewer 1r9r**
> > > > >
> > > > > >**[Experiments and Metrics Clarifications]:** The experimental section and method descriptions aren’t well aligned. For example, how do Negative Segment Referring, multi-turn dialogues, TABot (Video), TABot (Ours), and TABot-Chat (Ours) differ? How are metrics and tasks like Image Understanding related?
> > > > >
> > > > > **Answer:**
> > > > >
> > > > > (I) *Negative Segment Referring & Video Spatial Alignment in Table 5*: As mentioned earlier, these are the training recipes we proposed in designing our data paradigm. To validate the effectiveness of this new paradigm, we conducted related ablation studies by removing these two components one by one, as shown in Table 5.
> > > > >
> > > > > (ii) *Variants of TABot*:
> > > > > TABot (**Video**): **Stage-1** model and trained only with **video** data.
> > > > >
> > > > > TABot (**Image**): **Stage-1** model and trained only with **image** data.
> > > > >
> > > > > TABot (**Ours**): **Stage-1** model and trained with **video & image** data.
> > > > >
> > > > > TABot-**Chat** (Ours): **Stage-2** model and trained with **video & image** data.
> > > > >
> > > > > Multi-turn Dialogues: The multi-turn dialogues mentioned here refer specifically to the data used for instruction tuning. During functional tuning, all the data is single-turn. It is only in the instruction tuning stage that multi-turn dialogues are introduced.
> > > > >
> > > > > (iii) *Tasks and Metrics in Video Understanding*:
> > > > >
> > > > > 1. **Video Accident Recognition**:Similar to the image-based task, this is a binary classification task that determines whether a given video contains a traffic accident. The evaluation metrics are the same as those used for the image task.
> > > > > 2. **Video Accident Description**:This task evaluates whether the model-generated accident descriptions match the ground truth. The evaluation metrics are identical to those used for the image-based description task.
> > > > > 3. **Video Temporal Localization**:The model predicts the start and end time coordinates of a traffic accident on a normalized 1D timeline. We use IoU to measure the difference between the predicted temporal boundaries and the ground truth, reporting mIoU as the main metric. Additionally, we calculate AP@30, AP@50, and AP@70 in the same way as in the image spatial grounding task.
> > > > >
> > > > > (iv) *Tasks and Metrics in Image Understanding*:
> > > > >
> > > > > 1. **Image Accident Recognition**: This is a binary classification task that determines whether a given image contains a traffic accident. The evaluation metrics include overall accuracy as well as class-wise recall, precision, and F1-score for positive (accident) and negative (normal) samples.
> > > > > 2. **Image Accident Description**: This task evaluates whether the model-generated accident description accurately matches the ground truth. The evaluation metrics include BLEU-1 score, ROUGE-L F1 score, and BERT F1 score to measure the similarity between the predicted sentences and the ground truth. Additionally, we leverage GPT-4-Turbo to assess the alignment between the predictions and the ground truth.
> > > > > 3. **Image Spatial Grounding**: The model outputs bounding boxes to localize traffic accidents and related objects within the image. The evaluation metrics include mIoU. Furthermore, we consider a target to be successfully detected if its IoU exceeds 30, 50, and 70, respectively, and report AP@30, AP@50, and AP@70 to evaluate the model’s detection success rate under different IoU thresholds.
> > > > >
> > > > > ---
> > > > > **Additional responses are left to the next comment window.**

---

> ### Author Response · Authors · 2024-11-24
> **Response to Reviewer 1r9r**
>
> >**[Comparison on Other Models]:** The authors achieved remarkable results with GroundingGPT. How is the performance of the proposed method based on other models?
>
> **Answer:**
>
> We extend our experiments to include additional MLLMs fine-tuned on our proposed TAU-106K dataset, to highlight the significance and quality of our dataset.
> In detail, the very recent MLLMs, Video-LLaMA-2 and Qwen2-VL, are selected as the alternative baselines for comparison.
> Here we report the results of accident understanding tasks after fine-tuning the TAU-106K dataset in the following table:
>
> | Model         | V-CLS (Acc) | V-TL (AP@50) | V-CAP (BERT) | I-CLS (Acc) | I-AG (AP@50) | I-CAP (BERT) |
> | ------------- | ----------- | ------------ | ------------ | ----------- | ------------ | ------------ |
> | GroundingGPT  | 50.0        | 2.4          | 55.7         | 63.8        | 49.1         | 45.0         |
> | + TABot       | 81.0        | 20.1         | 82.3         | 90.8        | 65.9         | 75.2         |
> | Video-Llama-2 | 64.0        | 2.1          | 62.2         | 63.3        | 31.6         | 63.2         |
> | + TABot       | 79.9        | 34.3         | 83.4         | 77.8        | 57.3         | 73.7         |
> | Qwen2-VL      | 72.7        | 15.8         | 61.6         | 58.4        | 47.5         | 66.1         |
> | + TABot       | 82.7        | 22.5         | 83.1         | 92.0        | 77.6         | 76.3         |
>
> When fine-tuned on our TAU-106K dataset, the Video-LLaMA-2 and Qwen2-VL models achieve a significant improvement in all tasks, demonstrating the effectiveness of our proposed training recipes and the quality of our dataset.
> Notably, the CAP performance of Video-LLaMA-2 is even better than our proposed TABot model, and we conjecture that the performance gap is due to the enhanced captioning capabilities of Video-LLaMA-2 pre-trained on a larger-scale general-purpose dataset.
> After fine-tuning, the performance gap between Video-LLaMA-2 and GroundingGPT is narrowed, which indicates the importance of fine-tuning on the target dataset and the specificity of accident understanding tasks.
> In particular, the overall performance of our proposed TABot model based on GroundingGPT is still superior to fine-tuned Video-LLaMA-2, especially in image understanding tasks.
> As for the SOTA model Qwen2-VL, the pre-trained model already achieves competitive performance, and the fine-tuning on TAU-106K further boosts the performance, reaching the highest performance in all tasks.
> We conjecture that the competitive performance of the pre-trained Qwen2-VL is due to its advanced architecture and pre-training strategies, and maybe some accident-related or traffic-related data in the pre-training dataset.
> Even so, the performance improvement after fine-tuning on TAU-106K demonstrates the necessity of fine-tuning on the domain-specific dataset, highlighting the effectiveness and quality of our comprehensive TAU-106K dataset.
>
> ---
>
> >**[Performance Analysis and Discrepancy in Metrics]:** GroundingGPT achieves 100% accuracy on some metrics and 0% on others. What causes this disparity, and is it related to experimental settings or metric choices?
>
> **Answer:**
> First, we sincerely apologize for mistakenly swapping the Rec@A and Pre@A results for VTimeLLM and GroundingGPT in our original submission. In fact, both models achieve 100% Rec@A and 50% Pre@A. Here, "Rec" refers to recall, "Pre" refers to precision, "@A" represents class-wise results on the anomaly class, and "@N" represents class-wise results on the normal class.
>
> The reason for this error is that both GroundingGPT and VTimeLLM incorrectly classified all video data as containing a traffic accident. This leads to a recall of 100% on the anomaly class, but all metrics on the normal class drop to 0%. This indicates that neither GroundingGPT nor VTimeLLM can distinguish whether a traffic anomaly has occurred, which empirically proves the necessity of our proposed TABot model.
>
> ---
>
> >**[Case Study and Ground Truth Comparison]:** In the case study, how do prediction results from different methods compare to ground-truth annotations? Additionally, the conclusion mentions an image count of 55k, which seems inconsistent with earlier descriptions.
>
> **Answer:**
> We have added the ground truth in the section of the case study, and we can see that the output of our TABot is the closest to the ground truth.
>
> As for the image count of 55k mentioned in the conclusion, we apologize for the confusion. The precise number of images in our dataset is 54,767, as stated in the supplementary material, which is rounded to 55k in the conclusion.
> We will correct this inconsistency by updating `55k` to `54.8k` in the final version of the paper to ensure clarity and accuracy.
> Similarly, the detailed number of videos in the dataset is 51,544, which will be rounded to `51.5k` instead of `52k` in the final version.
> This change will not affect the overall conclusions or the name of the dataset, TAU-106K.

---

> ### Author Response · Authors · 2024-11-30
> **Dear Reviewer 1r9r**
>
> Dear **Reviewer 1r9r**,
>
> We would greatly appreciate it if you could review our response by **Dec 1 AoE**. After that date, it might be challenging for us to engage in further discussions.
>
> If you have any follow-up questions, please don't hesitate to reach out. We deeply value your expertise and time.
>
> Best regards,
> The authors of *TAU-106K: A New Dataset for Comprehensive Understanding of Traffic Accident*

---

### Official Review · Reviewer_oM9T · 2024-11-02

**Soundness:** 3
**Presentation:** 3
**Contribution:** 3
**Rating:** 8
**Confidence:** 5

**Summary:**

The authors introduce a novel dataset, TAU106k, designed to facilitate the training of multimodal Large Language Models (MLLMs) for a more comprehensive understanding of accidents. This dataset supports tasks such as accident recognition, temporal accident localization, and spatial-temporal object grounding. Additionally, the authors propose a method called TABot and compare its performance with various MLLMs on the TAU106k dataset.

**Strengths:**

1. **Traffic accident understanding is a critical task in the development of automated driving systems.** To support the training of multimodal Large Language Models (MLLMs) for traffic accident understanding, the authors have collected and annotated a large-scale dataset, TAU106k. This dataset includes structured annotations for accident recognition, temporal accident localization, and spatial-temporal object grounding. Overall, TAU106k comprises 52K video clips and 54K images.

2. **Training Grounding GPT-7B on this dataset yields substantial performance improvements across multiple tasks,** demonstrating that TAU106k effectively supports domain-specific applications.

**Weaknesses:**

**While the paper demonstrates significant effort in creating the dataset, I have the following concerns:**

1.	**Lack of clear motivation:** In the introduction, the authors do not clearly articulate the motivation for applying MLLMs to traffic accident understanding, especially since they begin by discussing “traffic accident detection.” Although MLLMs are attracting significant attention in the community, the value of using MLLMs, which may not necessarily operate in real-time, is unclear in this context. Justifying the need for this approach would strengthen the paper. In the rebuttal, I would like to hear the authors' feedback on this concern.

2.	**Limited comparison with existing datasets:** The authors are encouraged to provide a more comprehensive comparison between TAU106k and other existing datasets to highlight its unique contributions. For example, please demonstrate key features such as size, diversity of scenarios, types of annotations, or specific tasks supported.

3.	**Potential copyright issues with the dataset:** In line 191, the authors mention that the dataset is compiled from various sources, including academic datasets, YouTube, and BiliBili. While they have promised to release the dataset after acceptance, I am concerned that some of the collected data, particularly from platforms like YouTube, may involve copyright issues. If this is the case, the dataset and model release may not be feasible as promised, which would weaken the manuscript’s contribution. Please provide more details on your data collection process, including any steps taken to address copyright concerns, such as obtaining permissions or using only publicly licensed content

4.	**Annotator consistency and accuracy:** In lines 233-234, the authors state that their annotation strategy ensures consistency and coherence. However, it is unclear how many annotators are assigned to each item and how they ensure labeling accuracy if only one annotator is involved. Please provide the details on the annotation process, such as the number of annotators per item, any inter-annotator agreement measures used, and quality control procedures implemented.

5.	**Lack of a reasoning task:** In line 219, the authors note that “annotators are encouraged to infer the potential causes of the accident,” but this statement is vague. While the dataset could enable reasoning tasks for traffic accident understanding, the authors do not provide a detailed description or corresponding experiments. This omission detracts from the dataset’s uniqueness.

6.	**Limited improvement from the proposed algorithm:** In Tables 5 and 6, the proposed components show only incremental improvements over the base MLLM (GroundingGPT-7B). If the authors conducted cross-validation or multiple runs, these improvements might not be statistically significant, which could diminish the contribution of the proposed method.

Comments:

1. In L191, what is the method used from scene change detection?
2. L214, there is a typo (”Due”).

**Questions:**

1. Please clarify the motivation behind this work (Weakness #1).
2. Please address the copyright issues (Weakness #3).
3. Please explain how consistency in annotation is maintained (Weakness #4).
4. Please provide a rationale for the absence of a reasoning task in the experiments (Weakness #5).
5. Please discuss the limited performance improvement of the proposed algorithm (Weakness #6).

**Details Of Ethics Concerns:**

In line 191, the authors mention that the dataset is compiled from various sources, including academic datasets, YouTube, and BiliBili. While they have promised to release the dataset after acceptance, I am concerned that some of the collected data, particularly from platforms like YouTube, may involve copyright issues. If this is the case, the dataset and model release may not be feasible as promised, which would weaken the manuscript’s contribution.

---

> ### Author Response · Authors · 2024-11-24
> **Response to Reviewer oM9T**
>
> >**[Clear Motivation for Applying MLLMs]:** In the introduction, the authors do not clearly articulate the motivation for applying MLLMs to traffic accident understanding. The value of using MLLMs, which may not necessarily operate in real time, is unclear in this context.
>
> **Answer:**
>
> We will revise the introduction to explicitly highlight the advantages of MLLMs for traffic accident understanding, focusing on their ability to integrate multimodal data, provide contextual reasoning, and support downstream tasks such as accident causality analysis.
>
> 1. Comparison with Traditional Deep Learning Models: As stated in the introduction, traditional deep learning-based Traffic Accident Detection (TAD) methods often rely on conventional visual feature extraction techniques that lack high-level comprehension and reasoning capabilities. These models struggle with abstract concepts such as anomaly detection and interpreting intricate spatial-temporal dynamics in traffic accidents.
> The introduction of language models in MLLMs brings powerful reasoning capabilities, enabling them to bridge the above-mentioned gap by offering better semantic understanding and contextual reasoning capabilities for fine-grained accident analysis.
> Besides, MLLMs excel at aligning visual and textual modalities, enabling them to integrate multiple complex tasks (detect accidents, provide human-readable textual descriptions, identify accident boundaries in videos, and highlight accident-involved objects in images) in one model. As demonstrated in our experiments, these features significantly enhance the interpretability and utility of the system compared to traditional models.
>
> 1. Practical Applications Beyond Real-Time Scenarios: The use of MLLMs is not limited to real-time applications. The primary motivation for applying MLLMs is to advance the comprehension of traffic accidents, encompassing tasks such as accident reporting, post-incident analysis, training autonomous systems, illegal driving behavior analysis, insurance claims processing, driver education, and traffic safety research. These applications shall benefit from the high-level semantic understanding and detailed descriptions generated by MLLMs, which do not require real-time detection systems. Additionally, the structured and detailed accident annotations provided by TAU-106K enable the development of lightweight models that could inherit the semantic capabilities of MLLMs while being optimized for real-time deployment.
>
> ---
>
> **Additional responses are left to the next comment window.**

---

> > ### Author Response · Authors · 2024-11-24
> > **Response to Reviewer oM9T**
> >
> > >**[Comparison with Existing Datasets]:** The authors are encouraged to provide a more comprehensive comparison between TAU-106K and other datasets, focusing on key features such as size, diversity of scenarios, annotation types, or supported tasks.
> >
> > **Answer:**
> > Here we provide a more comprehensive comparison between TAU-106K and other datasets, focusing on key features such as size, domain, annotation types, and the characteristics of the textual captions. Here we report the comparison of TAU-106K with other accident-specific or general-purpose benchmarks in the following table:
> >
> >
> > | Dataset              | Years | Domain           | # Videos | Annotations                                                                 | # Words per caption | Avg. Duration |
> > | -------------------- | ----- | ---------------- | --------- | --------------------------------------------------------------------------- | -------------------- | ------------- |
> > | Dashcam              | 2016  | Traffic          | 3,000     | Temporal;                                                                   | -                    | 5.0 seconds   |
> > | A3D                  | 2019  | Traffic          | 1,500     | Temporal;                                                                   | -                    | 8.5 seconds   |
> > | CCD                  | 2021  | Traffic          | 1,500     | Temporal;                                                                   | -                    | 5.0 seconds   |
> > | TAD                  | 2021  | Traffic          | 500       | Accident Categories; Temporal;                                              | -                    | 35.8 seconds  |
> > | DADA                 | 2021  | Traffic          | 200       | Accident Categories; Temporal; Driver Attention                             | -                    | 11.0 seconds  |
> > | SUTD-TrafficQA       | 2021  | Traffic          | 10,080    | QA pairs                                                                    | -                    | 13.6 seconds  |
> > | DoTA                 | 2022  | Traffic          | 4,677     | Accident Categories; Temporal; Accident-involved Objects                    | -                    | 15.6 seconds  |
> > | CAP                  | 2023  | Traffic          | 11,727    | Accident Categories; Temporal; Fixed-Form Caption                           | 6.3 words            | 6.2 seconds   |
> > | TAD-106K             | 2024  | Traffic          | 51,544    | Accident Categories; Temporal; Accident-involved Objects; Free-Form Caption | 32.1 words           | 10.3 seconds  |
> > | Charades-STA         | 2017  | Daily activities | 9,848     | Temporal; Free-Form Caption                                                 | 6.3 words            | 31 seconds   |
> > | DiDeMo               | 2017  | Open             | 10,464    | Temporal; Free-Form Caption                                                 | 7.5 words            | 30 seconds   |
> > | ActivityNet-Captions | 2017  | Open             | 19,209    | Temporal; Free-Form Caption                                                 | 13.5 words           | 180 seconds  |
> >
> > According to the comparison, our TAU-106K is the largest dataset in terms of the number of videos and the variety of annotations, supporting a wide range of tasks including accident detection, temporal localization, accident grounding, and accident comprehension.
> > In particular, benefiting from our manual annotation process that is labor-intensive yet worthy, the labeled free-form accident captions in TAU-106K are much more diverse and detailed than other datasets, achieving a largest average length of 32.1 words per caption. This makes TAU-106K a valuable resource for training and evaluating accident-aware models in traffic video understanding.
> >
> > ---
> >
> > >**[Dataset Copyright Issues]:** There are concerns about copyright issues with data sourced from platforms like YouTube and BiliBili, which may weaken the manuscript’s contribution if data release is infeasible.
> >
> > **Answer:**
> > Our data source is from open-source academic benchmarks or public platforms. To avoid copyright and face privacy issues, we only provide video URLs and annotation data to the corresponding raw videos. To facilitate data download, we will also provide an open-source toolkit to access and download the related videos. This is a common practice in prior literature when publishing multimedia datasets, e.g. Youtube8M, HD-VG-130M, and Panda70M.
> >
> > ---
> >
> > **Additional responses are left to the next comment window.**

---

> > > ### Author Response · Authors · 2024-11-24
> > > **Response to Reviewer oM9T**
> > >
> > > >**[Annotator Consistency and Accuracy]:** Lines 233-234 mention that the annotation strategy ensures consistency and coherence, but details are lacking regarding the number of annotators, inter-annotator agreement measures, and quality control procedures.
> > >
> > > **Answer:**
> > >
> > > We use an internal annotation tool to enable interactive use with our annotaters and a diagram of the annotation protocol used in our data engine, which is illustrated in the **Figure 8 (Page 14) of the upgraded supplementary material PDF file**. Additionally, we provide detailed illustrations of the annotations for each accident types in the dataset, which are shown in the **Figure 9 (Page 15) of the upgraded supplementary material PDF file**.
> > >
> > > The annotation process is divided into two parts: video-based annotation and image-based annotation. Each part is separated into annotation and verification phases carried out by different annotators. The annotators in the verification phases are tasked with verifying the quality of each data item as "satisfactory" or "unsatisfactory". Unsatisfactory items were sent back to the annotation pipeline for refinement.
> > >
> > > Specifically, annotator A in the video annotation phase adopts Steps 1, 2 and 3 to provide the timestamps, semantic tags and detailed description for accident video, and another annotator B focuses on quality verification. After the video annotation, the fine video data is sent to the image annotation phase, annotator C utilizes Steps 1 and 2 to perform keyframe selection and accident-related object annotation, and then annotator D conducts the image-level quality verification. We employed a team of 50 experienced annotators and all of them followed the same annotation guidelines presented in both video and image verification phases. According to our annotation workflow, each data item involved at least four different annotators to uphold a high standard for annotation.
> > >
> > > Moreover, benefiting from our proposed video-to-image annotation pipeline, the image annotators and verifiers double-check the annotations from the video phase, ensuring the consistency and accuracy of the annotations across different modalities, which is also label-efficient and cost-effective.
> > >
> > > ---
> > >
> > > >**[Reasoning Tasks in Dataset]:** The dataset could enable reasoning tasks, but the description and experiments for this are missing, which detracts from its uniqueness.
> > >
> > > **Answer:** In the application of MLLMs to traffic accident understanding, the most critical task is to achieve precise temporal localization of accidents in videos.
> > > The labeled reason caption is a portent of the content of the accident, which makes accident detection and localization more trackable. Here we added an experiment to show the effectiveness of the reasoning caption on the temporal localization task:
> > >
> > > | Model            | AP@30 | AP@50 | AP@70 | mIOU  |
> > > | ---------------- | ----- | ----- | ----- | ----- |
> > > | TABot            | 39.44 | 20.12 | 9.80  | 25.93 |
> > > | - Reason Caption | 34.20 | 16.90 | 6.60  | 21.67 |
> > >
> > > As shown in the table, the removal of reasoning captions leads to a significant performance drop (4.26% on mIoU) on the temporal localization task, validating our claim that reasoning captions serve as valuable cues for accident understanding. And our future work will focus on developing more reasoning tasks based on the reasoning captions in TAU-106K, to achieve the accident forecasting and causality analysis tasks.
> > >
> > > ---
> > >
> > > >**[Incremental Algorithmic Improvement]:** The proposed components show only incremental improvements over GroundingGPT-7B, which might not be statistically significant.
> > >
> > > **Answer:** In our work, the TAU-106K dataset is the primary contribution, while the proposed components are designed to demonstrate the dataset’s utility and facilitate future research.
> > > On the other hand, our proposed training recipes (Video Spatial Alignment and Negative Segment Referring) have shown consistent improvements over the baseline model, indicating their effectiveness in enhancing accident-aware video understanding.
> > >
> > > ---
> > >
> > > >**[Scene Change Detection Method]:** In line 191, what is the method used for scene change detection?
> > >
> > > **Answer:**
> > > The widely-used toolkit PySceneDetect (https://www.scenedetect.com/) is employed for scene change detection, which is based on frame-level feature extraction and thresholding to identify abrupt changes in video content. In particular, `AdaptiveDetector` is employed in our video preprocessing pipeline.
> > >
> > > ---
> > >
> > > >**[Typographical Errors]:** Line 214 has a typo (“Due”).
> > >
> > > **Answer:** We will thoroughly review the manuscript to identify and correct typographical errors, ensuring that the text is polished and error-free.

---

> ### Author Response · Authors · 2024-11-30
> **Dear Reviewer oM9T**
>
> Dear **Reviewer oM9T**,
>
> We would greatly appreciate it if you could review our response by **Dec 1 AoE**. After that date, it might be challenging for us to engage in further discussions.
>
> If you have any follow-up questions, please don't hesitate to reach out. We deeply value your expertise and time.
>
> Best regards,
> The authors of *TAU-106K: A New Dataset for Comprehensive Understanding of Traffic Accident*

---

> > ### Comment · Reviewer_oM9T · 2024-12-02
> > **Thank you for the detailed response. My concerns are addressed.**
> >
> > I appreciate the authors spending their time addressing my concerns. My concerns are addressed in the rebuttal. Please incorporate all the additional results and discussions in the updated manuscript. I am happy to update my final score.

---

> > > ### Author Response · Authors · 2024-12-02
> > > **Thanks for your acknowledgement.**
> > >
> > > Dear **Reviewer oM9T**:
> > >
> > > Thank you for taking the time to review our rebuttal and for acknowledging our efforts in addressing your concerns. We greatly appreciate your constructive feedback and are pleased that our responses addressed your concerns effectively.
> > >
> > > We will **incorporate all the additional results and discussions from the rebuttal into the updated manuscript**, as per your suggestion. Your willingness to update the final score motivates us to further enhance the quality of our work. Thank you once again for your valuable time and effort in reviewing our paper.
> > >
> > >
> > > Best regards,
> > >
> > > The authors of *TAU-106K: A New Dataset for Comprehensive Understanding of Traffic Accident*

---

### Official Review · Reviewer_bWpe · 2024-11-04

**Soundness:** 3
**Presentation:** 3
**Contribution:** 4
**Rating:** 6
**Confidence:** 4

**Summary:**

This paper is focused on traffic accident analysis. The authors first introduced the TAU-106K dataset which contains both video and image modalities, and a wide range of fine-grained annotations including both spatial and temporal information. Then trained and evaluated a MLLM model, TABot. The training process has two stages, functional tuning and instruction tuning. The functional tuning involves multiple LLM training tasks to improve model performance in spatial and temporal understanding. The instruction tuning uses multi-round dialogue data generated based on captions by GPT-4o to learn the cause of accidents or prevention suggestions, which leads to the TABot-Chat model. TABot and TABot-Chat show superior performance in comprehensive understanding of traffic accident scenarios.

**Strengths:**

- Accident data as the rare cases are hard to collect, TAU-106K dataset provides 56% accident data and 44% normal which is a good balance overall. It has both video and image modalities with well annotated information from accident category, temporal location and description which is good for the comprehensive analysis of the accident videos from the ego car view.

- The majority of TAU-106K data is high quality as 720p, as image and video both are provided, it enables image-wise spatial annotation and training with that spatial information. There are also evaluation and ablation studies of (i) image-only model vs normal model, (ii) image task and video task, which is good insight for the community.

- The authors have annotators to precisely identify start and end timestamps of accidents within videos. This metadata is important in traffic scene analysis because it can distinguish the accident frames from non-accident frames, and also helps the Negative Segment Referring (NSR) task.

- The traffic accident description includes the aftermath, to help comprehend traffic accidents, e.g. potential causes of the accident.

-  Proposed method TABot follows the mainstream manner of training multi-modal LLM with a Q-former architecture. But it is good enough to test the different downstream tasks.

- Evaluation is done by comparison of the existing methods on the proposed dataset showing the challenge on the current accident localization.

**Weaknesses:**

- Comparison method for OSS model, which is not trained or fine-tuned on the same  train dataset that is hard to convince the the proposal method advantages.

- Accident data is not balanced for the fine-grained categories, e.g., MMV accidents take the major distribution. It is a concern for the training bias, this needs to be considered in the model design to avoid the data distribution affection for the inference performance. While, accidents and normal still balanced good.

- The reason behind the conclusion regarding the “coverage of the collected traffic data is good” needs to be supported more. As the paper focuses on traffic accident analysis and creating traffic accident dataset, it would be great to share more insights/data/analysis regarding this (e.g., list common traffic types/objects/scenarios and show that the dataset has covered them).

- Lack of comparison with the existing dataset regarding traffic accidents. It should have a single section and related work to describe it, currently the it is mixed in the related works, also lack of the comparison with some literatures such as :

WTS: A Pedestrian-Centric Traffic Video Dataset for Fine-grained Spatial-Temporal Understanding

**Questions:**

- TAU-106K is used as both the training set (=90% of the data) of the proposed TABot model and the testing set (=10% of the data) which is unbalanced. It is better to give more train/test splits tests, such as 7:3 for checking the model is not only benefit from the overfitting learning.

**Details Of Ethics Concerns:**

data are collected from youtube and bilibili need to take care about the human subjects privacy.

---

> ### Author Response · Authors · 2024-11-24
> **Response to Reviewer bWpe**
>
> > **[Fine-tuning on Other Baselines]:** Comparison method for the OSS model, which is not trained or fine-tuned on the same train dataset is hard to convince the proposal method advantages.
>
> **Answer:**
>
> We extend our experiments to include additional MLLMs fine-tuned on our proposed TAU-106K dataset, to highlight the significance and quality of our dataset.
> In detail, the very recent MLLMs, Video-LLaMA-2 and Qwen2-VL, are selected as the alternative baselines for comparison.
> Here we report the results of accident understanding tasks after fine-tuning the TAU-106K dataset in the following table:
>
> | Model         | V-CLS (Acc) | V-TL (AP@50) | V-CAP (BERT) | I-CLS (Acc) | I-AG (AP@50) | I-CAP (BERT) |
> | ------------- | ----------- | ------------ | ------------ | ----------- | ------------ | ------------ |
> | GroundingGPT  | 50.0        | 2.4          | 55.7         | 63.8        | 49.1         | 45.0         |
> | + TABot       | 81.0        | 20.1         | 82.3         | 90.8        | 65.9         | 75.2         |
> | Video-Llama-2 | 64.0        | 2.1          | 62.2         | 63.3        | 31.6         | 63.2         |
> | + TABot       | 79.9        | 34.3         | 83.4         | 77.8        | 57.3         | 73.7         |
> | Qwen2-VL      | 72.7        | 15.8         | 61.6         | 58.4        | 47.5         | 66.1         |
> | + TABot       | 82.7        | 22.5         | 83.1         | 92.0        | 77.6         | 76.3         |
>
> When fine-tuned on our TAU-106K dataset, the Video-LLaMA-2 and Qwen2-VL models achieve a significant improvement in all tasks, demonstrating the effectiveness of our proposed training recipes and the quality of our dataset.
> Notably, the CAP performance of Video-LLaMA-2 is even better than our proposed TABot model, and we conjecture that the performance gap is due to the enhanced captioning capabilities of Video-LLaMA-2 pre-trained on a larger-scale general-purpose dataset.
> After fine-tuning, the performance gap between Video-LLaMA-2 and GroundingGPT is narrowed, which indicates the importance of fine-tuning on the target dataset and the specificity of accident understanding tasks.
> In particular, the overall performance of our proposed TABot model based on GroundingGPT is still superior to fine-tuned Video-LLaMA-2, especially in image understanding tasks.
> As for the SOTA model Qwen2-VL, the pre-trained model already achieves competitive performance, and the fine-tuning on TAU-106K further boosts the performance, reaching the highest performance in all tasks.
> We conjecture that the competitive performance of the pre-trained Qwen2-VL is due to its advanced architecture and pre-training strategies, and maybe some accident-related or traffic-related data in the pre-training dataset.
> Even so, the performance improvement after fine-tuning on TAU-106K demonstrates the necessity of fine-tuning on the domain-specific dataset, highlighting the effectiveness and quality of our comprehensive TAU-106K dataset.
>
> ---
>
>
>
> > **[Data Imbalance in Fine-grained Categories]:** Accident data is not balanced for the fine-grained categories, e.g., MMV accidents take the major distribution. This raises concerns about training bias affecting inference performance.
>
> **Answer:**
> The class imbalance issue is inevitable in data collection and our data distribution also fairly reflects real-world situations. The primary purpose of this paper is to facilitate the development of MLLM on large-scale traffic dataset and learn models that closely resemble real-world conditions. Besides, we provide a table showing the accuracy of different types of traffic accidents in both video/image accident recognition tasks. While categories like MnMV show slightly lower performance due to the limited amount of training data, the model's performances across different accident types are generally comparable. Overall, the model achieves consistent and satisfactory results across the normal class and the five accident classes.
>
> | Accident Category | Proportion | Acc@Video | Acc@Image |
> | ----------------- | ---------- | --------- | --------- |
> | Normal            | 44.1%      | 78.80     | 92.23     |
> | SMV               | 9.7%       | 80.56     | 79.75     |
> | MMV               | 32.9%      | 84.39     | 87.40     |
> | MV&nMV            | 9.8%       | 92.02     | 82.87     |
> | V&P               | 2.2%       | 94.29     | 84.78     |
> | MnMV              | 1.3%       | 77.78     | 78.26     |
>
> ---
>
> **Additional responses are left to the next comment window.**

---

> > ### Author Response · Authors · 2024-11-24
> > **Response to Reviewer bWpe**
> >
> > >**[Dataset Coverage Analysis]:** The conclusion that “coverage of the collected traffic data is good” needs more support. It would be helpful to list common traffic types/objects/scenarios and demonstrate that the dataset covers them adequately.
> >
> > **Answer:**
> > Following the reviewer's suggestion, we have conducted a detailed analysis of the dataset coverage to demonstrate its diversity and representativeness.
> > In detail, our dataset covers a wide range of traffic scenarios, objects, and accident categories:
> > - **Traffic Scenarios:** urban streets (49%), intersections (19%), country roads (17%), highways (12%), and other traffic scenes (3%) like parking lots, pedestrian crossings, gas station, tunnel, and roundabout, etc.
> > - **Objects:** cars (58%), trucks (12%), electric bikes (11%), pedestrians (5%), vans (3%), bicycles (3%), buses (2%), guardrails (2%), motorcycles (2%), and other objects (2%) like traffic lights, trees, and animals, etc.
> > - **Accident Categories:** 5 pre-defined categories: multi-motor-vehicle accidents (MMV, 59%), motor-vehicle & non-motor-vehicle accidents (MV&nMV, 18%), single-motor-vehicle accidents (SMV, 17%), vehicle & pedestrian accidents (V&P, 4%), and multi-non-motor-vehicle accidents (MnMV, 2%).
> >
> > ---
> >
> > >**[Comparison with Existing Datasets]:** Lack of comparison with existing datasets regarding traffic accidents. The paper should include a dedicated section for this, with comparisons to works like "WTS: A Pedestrian-Centric Traffic Video Dataset for Fine-grained Spatial-Temporal Understanding."
> >
> > **Answer:**
> > We will provide a more comprehensive comparison between TAU-106K and other datasets, focusing on key features such as size, domain, annotation types, and the characteristics of the textual captions. Here we report the comparison of TAU-106K with other accident-specific or general-purpose benchmarks in the following table:
> >
> > | Dataset              | Years | Domain           | # Videos | Annotations      | # Words per caption | Avg. Duration |
> > | -------------------- | ----- | ---------------- | --------- | --------------------------------------------------------------------------- | -------------------- | ------------- |
> > | Dashcam              | 2016  | Traffic          | 3,000     | Temporal;     | -                    | 5.0 seconds   |
> > | A3D                  | 2019  | Traffic          | 1,500     | Temporal;  | -                    | 8.5 seconds   |
> > | CCD                  | 2021  | Traffic          | 1,500     | Temporal;                                                                   | -                    | 5.0 seconds   |
> > | TAD                  | 2021  | Traffic          | 500       | Accident Categories; Temporal;                                              | -                    | 35.8 seconds  |
> > | DADA                 | 2021  | Traffic          | 200       | Accident Categories; Temporal; Driver Attention                             | -                    | 11.0 seconds  |
> > | SUTD-TrafficQA       | 2021  | Traffic          | 10,080    | QA pairs                                                                    | -                    | 13.6 seconds  |
> > | DoTA                 | 2022  | Traffic          | 4,677     | Accident Categories; Temporal; Accident-involved Objects                    | -                    | 15.6 seconds  |
> > | CAP                  | 2023  | Traffic          | 11,727    | Accident Categories; Temporal; Fixed-Form Caption                           | 6.3 words            | 6.2 seconds   |
> > | TAD-106K             | 2024  | Traffic          | 51,544    | Accident Categories; Temporal; Accident-involved Objects; Free-Form Caption | 32.1 words           | 10.3 seconds  |
> > | Charades-STA         | 2017  | Daily activities | 9,848     | Temporal; Free-Form Caption                                                 | 6.3 words            | 31 seconds   |
> > | DiDeMo               | 2017  | Open             | 10,464    | Temporal; Free-Form Caption                                                 | 7.5 words            | 30 seconds   |
> > | ActivityNet-Captions | 2017  | Open             | 19,209    | Temporal; Free-Form Caption                                                 | 13.5 words           | 180 seconds  |
> >
> >
> > According to the comparison, our TAU-106K is the largest dataset in terms of the number of videos and the variety of annotations, supporting a wide range of tasks including accident detection, temporal localization, accident grounding, and accident comprehension.
> > In particular, benefiting from our manual annotation process that is labor-intensive yet worthy, the labeled free-form accident captions in TAU-106K are much more diverse and detailed than other datasets, achieving a largest average length of 32.1 words per caption. This makes TAU-106K a valuable resource for training and evaluating accident-aware models in traffic video understanding.
> >
> > ---
> >
> > **Additional responses are left to the next comment window.**

---

> > > ### Author Response · Authors · 2024-11-24
> > > **Response to Reviewer bWpe**
> > >
> > > >**[Train/Test Splits for Validation]:** TAU-106K is used with a 90/10 train/test split, which is unbalanced. Alternative splits like 7:3 are suggested to verify the model doesn’t overly benefit from overfitting.
> > >
> > > **Answer:** Since the amount of our TAU-106K dataset is large enough, 1/10 of the data (5K videos and 5K images) is sufficient for testing.
> > > However, we have conducted additional experiments with a 7:3 train/test split to verify the model's generalization ability and robustness to overfitting. The results are reported in following table. The model's performance remains consistent across different tasks with a slight decrease in accuracy, demonstrating its robustness to different train/test splits.
> > >
> > > | Model         | V-CLS (Acc) | V-TL (AP@50) | V-CAP (BERT) | I-CLS (Acc) | I-AG (AP@50) | I-CAP (BERT) |
> > > | ------------- | ----------- | ------------ | ------------ | ----------- | ------------ | ------------ |
> > > | TABot (9:1)  | 81.00       | 20.12        | 82.31       | 90.75       | 70.03       | 75.20        |
> > > | TABot (7:3)  | 79.95       | 19.08        | 81.57       | 88.97       | 68.29       | 74.55        |
> > >
> > >
> > > ---
> > >
> > > >**[Ethics Concerns: Data Privacy]:** Data collected from YouTube and Bilibili may raise privacy concerns regarding human subjects.
> > >
> > > **Answer:** To avoid copyright and face privacy issues, we only provide video URLs and annotation data to the corresponding raw videos. To facilitate data download, we will also provide an open-source toolkit to access and download the related videos. This is a common practice in prior literature when publishing multimedia datasets, e.g. Youtube8M, HD-VG-130M, and Panda70M.

---

> ### Author Response · Authors · 2024-11-30
> **Dear Reviewer bWpe**
>
> Dear **Reviewer bWpe**,
>
> We would greatly appreciate it if you could review our response by **Dec 1 AoE**. After that date, it might be challenging for us to engage in further discussions.
>
> If you have any follow-up questions, please don't hesitate to reach out. We deeply value your expertise and time.
>
> Best regards,
> The authors of *TAU-106K: A New Dataset for Comprehensive Understanding of Traffic Accident*

---

> ### Comment · Reviewer_bWpe · 2024-12-02
> **Thanks for the PC and Reviewer Answer**
>
> Thanks for the PC to clarify the comments, and after reading the respones from the reviewer,
> it is almost clear for my concern, the added experiment regarding the additional MLLMs fine-tune result, class banlance analysis, and traffic situation coverage are orgnized well and better to be mentioned in the final manuscript,
> I would like to accept this paper and keep my rating according to the contribution of this paper.

---

> > ### Author Response · Authors · 2024-12-02
> > **Thanks for your Recognition**
> >
> > Dear Reviewer bWpe:
> >
> > Thank you for your valuable feedback and for carefully reviewing our rebuttal. We are pleased to know that the additional experiments and statements, including the fine-tuning results with additional MLLMs, class balance analysis, and traffic situation coverage, effectively addressed your concerns.
> >
> > As you suggested, we will ensure these results and discussions are clearly highlighted in the revised manuscript to further strengthen its contribution. Thank you once again for your support and recognition of our work.
> >
> > Best regards,
> >
> > The authors of *TAU-106K: A New Dataset for Comprehensive Understanding of Traffic Accident*

---

### Author Response · Authors · 2024-11-24
**Overall Response to Reviewers**

We sincerely thank all reviewers for acknowledging the significance and novelty of our work.
We are greatly encouraged by the reviewers' recognition of the following aspects:

1. The significance and comprehension of the proposed TAU-106K dataset for traffic accident understanding.
2. The effectiveness of the proposed training recipes and the improvements they bring to the baseline model.
3. The publication of the dataset and the code to facilitate future research in the field.

We have carefully considered all reviewers' comments and suggestions and summarized some common concerns about our dataset and methodology as follows:

**1. [Comparison with Existing Datasets]:** We provide a more comprehensive comparison between TAU-106K and other datasets, focusing on key features such as size, domain, annotation types, and the characteristics of the textual captions. Through the comparison, the unique advantages of TAU-106K in terms of the number of videos, the variety of annotations, and the diversity and flexibility of the free-form accident captions are highlighted.

**2. [Annotation Process and Quality Control]:** We provide a detailed description and illustration of the annotation process and quality control mechanism, including the annotation tool, the annotation protocol, and the verification phases. We also present the workflow of the video-to-image annotation pipeline, which ensures the consistency and accuracy of the annotations across different modalities.

**3. [Copyright Issues of Dataset]:** We clarify that the data source of TAU-106K is from open-source academic benchmarks or public platforms. To avoid copyright and privacy issues, we only provide video URLs and annotation data to the corresponding data sources.

**4. [Fine-tuning on Other Baselines]:** We extend our experiments to include additional MLLMs fine-tuned on our proposed TAU-106K dataset, to highlight the significance and quality of our dataset.

Besides these common concerns, we have also one-by-one addressed each reviewer's specific comments and suggestions in detail.

According to the reviewers' feedback, we have revised the manuscript and upload the **revision** to the platform, and extended the **supplementary material** to provide more detailed information about the dataset and the methodology.
We believe that the revised version has significantly improved the quality and clarity of the paper.

---

### Meta-Review · Area_Chair_AG3o · 2024-12-17

**Metareview:**

After discussion, this submission received 4 positive scores . The major concerns about `comparison with other datasets`, `annotation process and quality control`, `copyright issue` and `fine-tuning on other baselines` were comprehensively solved. After reading the paper, the review comments and the rebuttal, the AC thinks the remaining issue is to include all the revised content to the camera-ready version and correct typographical errors.

**Additional Comments On Reviewer Discussion:**

After discussion, all the reviewers gave the positive scores. Reviewer oM9T updated the score to 8 while Reviewer #1r9r and #Vxed keep the original ratings. The authors carefully responded to the questions and revised the paper according to the concerns from all the reviewers.

---

### Decision · Program_Chairs · 2025-01-22

Accept (Poster)